# A hybrid-grid global model for the estimation of atmospheric weighted mean temperature considering time-varying lapse rate in GNSS precipitable water vapor retrieval

Shaofeng Xie[1], Jihong Zhang[2], Liangke Huang[1], Fade Chen[1], Yongfeng Wu[1], Yijie Wang[3], Lilong Liu[1]

[1]College of Geomatics and Geoinformation, Guilin University of Technology, Guilin, 541004, China
[2]Faculty of Land and Resources Engineering, Kunming University of Science and Technology, Kunming, 650093, China
[3]Jiujiang Surveying and Mapping Geographic Information Co., Ltd., Jiujiang, 32005, China

*Correspondence to*: Jihong Zhang (jhzhang@stu.kust.edu.cn) and Liangke Huang (lkhuang@whu.edu.cn)

**Abstract.** The atmospheric weighted mean temperature ($T_m$) is a key parameter in global navigation satellite system (GNSS) water vapor retrieval and can convert the zenith wet delay (ZWD) into precipitable water vapor (PWV). However, there are some shortcomings in the existing $T_m$ models, such as the detailed time-varying lapse rate not being considered. Additionally, the spatiotemporal characteristics of $T_m$ need to be further refined. Therefore, we developed a new global high-precision and high-spatiotemporal-resolution $T_m$ model considering time-varying lapse rate using the latest European Centre for Medium-Range Weather Forecasts ReAnalysis 5 (ERA5) atmospheric reanalysis data. Firstly, a global multidimensional $T_m$ lapse rate model (NGGTm-H model) was developed using the sliding window algorithm. Secondly, the daily variation characteristics of $T_m$ and its relationships with geographical situation were investigated. Finally, a hybrid-grid global $T_m$ model considering time-varying lapse rate (NGGTm model) was developed. To verify the effectiveness of the proposed model, the NGGTm model was compared with the Bevis and GPT3 models using the $T_m$ data recorded at 378 radiosonde stations in 2017 and the surface grid $T_m$ data calculated from the ERA5 reanalysis data. The results show that taking the surface grid $T_m$ data of ERA5 as reference values, the average root mean square error (RMSE) value predicted by the NGGTm model was 2.84 K, which was higher with 0.50 K, 0.18 K and 0.06 K than those of the Bevis, GPT3-5 and GPT3-1 models, respectively. Meanwhile, taking the $T_m$ data from the radiosonde stations as the reference values, the mean bias and RMSE of the NGGTm model were 0.10 K and 3.30 K, respectively, which exhibit the best accuracy and stability among the Bevis, GPT3-5 and GPT3-1 models.

## 1 Introduction

Precipitable water vapor (PWV), a basic component of the water cycle of the Earth, is a key parameter in climate variation and material and energy exchange research performed at the global scale (Huang et al., 2023; Ding et al., 2022). PWV directly impacts the ground temperature and air humidity (Rocken et al., 1997). Furthermore, PWV is highly active in the Earth's atmosphere and plays a crucial role in the formation and evolution of weather. Its temporal and spatial variations are

essential for the development of clouds and rainfall (Philipona et al., 2005; Jin & Luo, 2009). Understanding the exact spatiotemporal features of global PWV variations holds immense practical importance for monitoring and forecasting catastrophic weather events and conducting research on climate change. However, atmospheric PWV is highly susceptible to the underlying terrain, seasonal variations, and other climate changes, causing its spatial distribution to change uneven and rapidly over time. Therefore, accurately monitoring PWV poses a significant challenge (Wang et al., 2007; Wang & Zhang, 2009). Currently, the methods for deriving PWV mainly include radiosonde, ground-based detection, microwave radiometer and satellite remote sensing inversion methods (Alexandrov et al., 2009; Gui et al., 2017; Zeng et al., 2019). Each technology has its own set of advantages and limitations. Radiosondes, for example, are highly accurate in measuring meteorological parameters but are limited by their low spatiotemporal resolution, high observation costs, and inability to provide real-time or near-real-time updates on PWV changes (Zhai & Eskridge, 1996). Microwave radiometers and satellite remote sensing, which rely on infrared band detection, offer high detection accuracies. However, their effectiveness is limited by interference from weather conditions such as clouds, fog, rain, and snow. Additionally, these instruments are unable to provide profile information of PWV in the vertical direction, and this shortcoming restricts their applicability in PWV detection tasks (Dalu, 1986; Gao & Kaufman, 2003).

Global Navigation Satellite System (GNSS) has become a crucial technology for real-time and high-precision PWV detection with advantages of all-weather capability, a high spatiotemporal resolution, low costs, and weather resistance (Zhao et al., 2018; Jiang et al., 2017; Manandhar et al., 2017; Huang et al., 2021; Huang et al., 2022). The precision of GNSS-derived PWV can be as high as 1 to 1.5 mm, with a temporal resolution of 0.5 hours (Rocken et al., 1993; Adams et al., 2011). The tropospheric delay can be expressed as the zenith total delay (ZTD), which consists of two parts: the zenith hydrostatic delay (ZHD) and zenith wet delay (ZWD). The ZTD is an important factor affecting high precision GNSS positioning and also the basic data for GNSS atmospheric research (Huang et al., 2023c; Zhu et al., 2022). According to the high-precision observation data provided by the GNSS base station network, high-precision ZTD information can be obtained through data processing with high-precision GNSS data processing software or by integrating atmospheric reanalysis data. The ZHD values, with strong variation regularity, can be calculated by a simple model using surface pressure data to obtain an accuracy at the millimeter level. However, the variation law of ZWD influenced mainly by water vapor is difficult to investigate (Vedel et al., 2001). The ZWD can be computed by subtracting the ZHD from the ZTD. Then, the result can be converted to PWV by using the water vapor conversion factor. Among the parameters involved, the atmospheric weighted mean temperature ($T_m$) is the key parameter for calculating the water vapor conversion factor. The accuracy of GNSS tropospheric water vapor retrievals can be significantly improved by using high-precision $T_m$ data.

High precision $T_m$ data can typically be calculated by integrating radiosonde data, atmospheric reanalysis data, and numerical weather prediction (NWP) data. However, the distribution of radiosonde stations is uneven, and there is a time delay in releasing atmospheric reanalysis data. In addition, NWP data are subject to certain limitations, including low temporal resolution and slow update speed, which renders them unsuitable for real-time or near-real-time PWV monitoring (Zhang et al., 2017). To improve the calculation efficiency of $T_m$, it is necessary to build a real-time and high-precision $T_m$

Open Access EGU

model to meet the needs of GNSS PWV inversion. Existing $T_m$ models can be divided into two categories: meteorological parameter models and nonmeteorological parameter models. By analyzing the correlation between the surface temperature ($T_s$) and $T_m$ and utilizing two-year profile information from 13 radiosonde stations in North America, the Bevis formula was developed through linear regression analysis (Bevis et al., 1992). This formula can successfully retrieve PWV information in the zenith direction of the station using GPS observation data and introduced the concept of GPS in meteorological research for the first time. The linear regression model remains a reliable and convenient tool that is still widely used today. However, it is important to note that the coefficients of this model exhibit distinct characteristics based on the region and season in which it is applied. Therefore, recalculating the parameters of the model is necessary when applying the model in other regions (Ross & Rosenfeld, 1997; Emardson et al., 1998). With the continuous development of GNSS PWV detection technology, many scholars have refined and expanded the Bevis model regionally and developed other $T_m$ models based on measured meteorological parameters. Besides $T_s$, $T_m$ is also related to $P_s$ and $e_s$. The global single-factor $T_m$ model and multifactor $T_m$ model were developed, which showed better accuracy and reliability (Yao et al., 2014c). To achieve better results in the global range, Yao et al. (2014b) proposed a $T_m$ linear regression model in each latitude interval region using the European Centre for Medium-Range Weather Forecasts (ECMWF) reanalysis data. In addition, neural network algorithm can be used to establish $T_m$ model that can output corresponding $T_m$ values by simply inputting $T_s$ information. The accuracy of this model is dependent on the precision of the input $T_s$ information. When highly precise $T_s$ data were used, the model accuracy was increased (Ding, 2018). The above models have achieved good results when providing the required measured meteorological parameters, but most of the GNSS stations in the world do not have supporting meteorological sensors installed leading to the difficulty for measuring meteorological parameters in real-time. Therefore, these models are difficult to apply to real-time or near-real-time GNSS PWV detection tasks. To realize real-time GNSS PWV detection, many scholars have developed $T_m$ models (empirical models) that run without measured meteorological parameters. For example, Zhu et al. (2021) developed a new $T_m$ model taking climate differences into account in the Shanxi region. The non-meteorological parameter $T_m$ model (named the Emardson model) was developed to take the annual cycle variation into account by using radiosonde data collected in Europe over many years, which was capable of meeting the requirement for GNSS water vapor detection (Emardson & Derks, 2000). Therefore, the model has been widely used in real-time GNSS meteorology research. Additionally, the lapse rate is the key parameter in the $T_m$ elevation correction. Taking the lapse rate into account can not only improve the $T_m$ model accuracy, but also showed significant performances in regions with undulating terrain (Huang et al., 2023b; Sun et al., 2021; Yao et al., 2018). The $T_m$ lapse rate is an effective means of not only correcting $T_m$ to different surface heights but also providing a vertical correction value for $T_m$ at any height. Therefore, investigating the spatiotemporal variation characteristics of the $T_m$ lapse rate and developing a $T_m$ lapse rate model have high application values in $T_m$ vertical and spatial interpolation tasks. Furthermore, a high-precision global $T_m$ model that considers elevation, latitude, and time in real time could greatly enhance the accuracy of GNSS PWV monitoring. Although the aforementioned models excel in certain regions and possess unique strengths, they are not suitable for calculating $T_m$ at

the global level. Yao et al. (2012) developed the first new global atmospheric weighted average temperature model (GWMT model) using data from 135 radiosonde stations worldwide over several years. This new model can estimate the $T_m$ value at any location by simply inputting the station location and the annual product day, which have been applied to real-time GNSS PWV inversion studies worldwide. However, because the radiosonde data used in the GWMT model are all located on land, there is a certain systematic bias in ocean areas. To address this issue, the GTm-II model, GTm-III model, and GTm-H model were developed jointing atmospheric reanalysis data (Yao et al., 2013; 2014a). GPT-series models also show excellent performance worldwide (Landskron & Böhm, 2018; Böhm et al., 2007; Böhm et al., 2015). Moreover, some scholars have improved the GPT2w model (Yang et al., 2020; Huang et al., 2019b). Although the GPT3 model is currently the most representative empirical model with a high precision on the global scale, GPT3 model dose not take into account elevation correction or detailed $T_m$ lapse rate. Thus, it is necessary to develop a new model to improve the real-time high-precision global empirical $T_m$ model and to select appropriate data sources for model development.

The global $T_m$ models mentioned above were established without accounting for the detailed time-varying lapse rate. Therefore, in this study, we aim to global $T_m$ model that takes into account time-varying lapse rate and high-precision capabilities. To attain this objective, firstly, we investigated the spatiotemporal variations and characteristics of the lapse rate of global $T_m$ and developed the lapse rate model (NGGTm-H). Secondly, a hybrid-grid global model (NGGTm) for the estimation of atmospheric weighted mean temperature considering time-varying lapse rate was developed by using profile gridded $T_m$ data calculated by integrating the latest European Centre for Medium-Range Weather Forecasts ReAnalysis 5 (ERA5) reanalysis data. To verify the effectiveness of the new model, the NGGTm model was compared with the Bevis and GPT3 models using $T_m$ data from radiosonde stations with ERA5 reanalysis data.

## 2 Data and methodology

### 2.1 Data

The ERA5 atmospheric reanalysis dataset, provided by ECMWF (https://apps.ecmwf.int/datasets/data/interim-full-daily), is the fifth-generation global climate reanalysis dataset. This dataset provides hourly surface-level parameters and pressure-level data with a horizontal resolution of 0.25°×0.25° (latitude×longitude) and a vertical resolution of 37 levels. ERA5 data can provide high-resolution and relatively complete surface-level and pressure-level data and are thus expected to be widely used in the future. The radiosonde station data can be downloaded for free from the University of Wyoming (http://weather.uwyo.edu/upperair/sounding.html). This product provides meteorological layered data and surficial parameters such as atmospheric water vapor from the ground to the near-Earth space (an altitude of approximately 30 km) and provides radiosonde data twice a day (UTC 00:00 and 12:00); these data are often used as reference values for model verification tasks. The ERA5 gridded data from 2012 to 2017 and the radiosonde data in 2017 on the global scale were used to analyze and develop the model in this study.

### 2.2 Methodology

$T_m$ is the key parameter used to convert ZWD into PWV. Using atmospheric reanalysis data, radiosonde data and other data, highly accurate $T_m$ information can be obtained by integral calculation. In addition, the modeling method can also obtain $T_m$ values at a high accuracy and with a high calculation efficiency. The specific $T_m$ integral calculation formula is expressed as follows:

$$T_m = \frac{\int (e/T)\, dH}{\int (e/T^2) dH} \tag{1}$$

where $e$ is the water vapor pressure (hPa), $T$ is the temperature (K), and $H$ is the integral range (m).

The modeling methods used to calculate $T_m$ can be divided into two categories: ① $T_m$ models based on measured meteorological parameters, of which the most representative is the Bevis model ($T_m$=70.2+0.72$T_s$), namely, the $T_m$ linear regression model, and ② non-meteorological parameter $T_m$ models, of which the most classical is the GPT series model. The GPT3 model, the latest model in the GPT series, has a high accuracy in calculating global $T_m$. The GPT3 model formula used to calculate $T_m$ can be expressed as follows:

$$T_m^{GPT3} = a_0 + a_1 \cos\left(2\pi \frac{DOY}{365.25}\right) + b_1 \sin\left(2\pi \frac{DOY}{365.25}\right) + a_2 \cos\left(4\pi \frac{DOY}{365.25}\right) + b_2 \sin\left(4\pi \frac{DOY}{365.25}\right) \tag{2}$$

where $T_m^{GPT3}$ denotes $T_m$ calculated by the GPT3 model, $a_0$ denotes the average annual value of $T_m$, $a_1$ and $b_1$ denote the annual cycle coefficient of $T_m$, $a_2$ and $b_2$ denote the semiannual cycle coefficient of $T_m$, and $DOY$ denotes the day of the year.

PWV refers to the total water vapor content of a vertical column per unit area in the atmosphere. PWV can be converted from ZWD using the following formula:

$$PWV = \Pi \times ZWD \tag{3}$$

where Π denotes the PWV conversion factor. This conversion factor can be expressed as follows:

$$\Pi = \frac{10^6}{\rho_w R_v (k_2' + k_3/T_m)} \tag{4}$$

where $R_v$ denotes the water vapor gas constant, $k_2'$ and $k_3$ are constants ($k_2'$=22.97 K/hPa and $k_3$=375463 $K^2$/hPa), and the other parameters are described above. Therefore, $T_m$ is the key parameter in the GNSS PWV inversion.

To facilitate the subsequent test of the accuracy of the $T_m$ values calculated using the atmospheric reanalysis data and the performance of the new $T_m$ model, this study uses the bias and root mean square error (RMSE) as the accuracy evaluation indicators, calculated using the following formulas:

$$bias = \frac{1}{N}\sum_{i=1}^{N}(KP_M^i - KP_R^i) \tag{5}$$

$$RMSE = \sqrt{\frac{1}{N}\sum_{i=1}^{N}(KP_M^i - KP_R^i)^2} \tag{6}$$

where $N$ denotes the number of samples, $KP_M^i$ denotes the calculated value of the atmospheric reanalysis data or model, and $KP_R^i$ denotes the reference value.

## 3 Development of the $T_m$ lapse rate model

### 3.1 Analysis of the spatiotemporal characteristics of the $T_m$ lapse rate

Given the discernible variations in topography and the significant range of elevation at a global scale, there can be considerable disparities between the atmospheric reanalysis data grid points and the actual elevations of the user. The variation in the $T_m$ elevation is much larger than the variation in the horizontal direction, so it is necessary to correct for the vertical $T_m$ information provided by the reanalysis data. The previous study has analyzed this topic and concluded that there is an approximately linear relationship between the layered gridded $T_m$ data and elevation (Huang et al., 2019a). To analyze the change in $T_m$ with elevation in depth, six representative ERA5 reanalysis data grid points ((60° N, 90° W), (60° N, 90° E), (0°, 90° W), (0°, 90° E), (60° S, 90° W) and (60° S, 90° E)) were selected globally to analyze the grid-level $T_m$ data and corresponding height data on January 1st, 2017. The results are shown in Fig. 1.

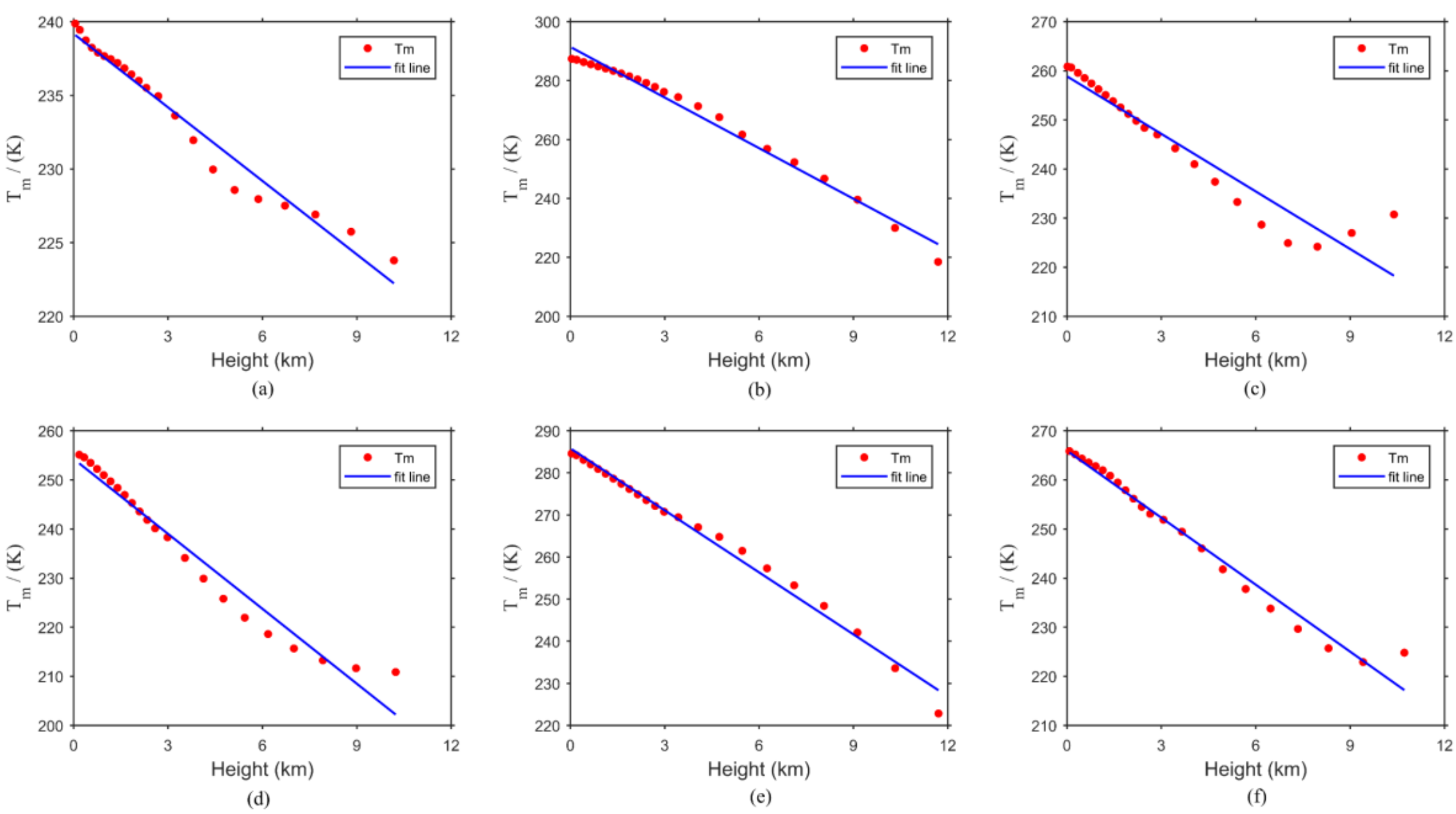

**Figure 1 $T_m$ changes with elevation at six representative ERA5 reanalysis data gridded points on January 1st, 2017. (a) (60° N, 90° W). (b) (0°, 90° W). (c) (60° S, 90° E). (d) (60° N, 90° E). (e) (0°, 90° E). (f) (60° S, 90° W).**

Figure 1 shows that the grid-level $T_m$ data of six representative ERA5 reanalysis data grid cells exhibit approximate linear change relationships with elevation. Moreover, the grid-level $T_m$ data gradually decrease with increasing elevation. Therefore, the slope of the fitting line represents the lapse rate of $T_m$, and this relation can be expressed as follows:

$$T_m = \gamma \times \delta h + l \tag{7}$$

where $\gamma$ denotes the lapse rate of $T_m$, $\delta h$ denotes the height, and $l$ denotes a constant.

To investigate the variation relationship between the lapse rate of $T_m$ and time at the global scale, six representative ERA5 reanalysis data grid points were selected to calculate the lapse rate of $T_m$ from 2012 to 2016. Furthermore, the time series for the lapse rate of the daily mean $T_m$ from 2012 to 2016 was obtained and used to achieve seasonal fitting by the cosine function of the annual and semiannual periods. The results are shown in Fig. 2.

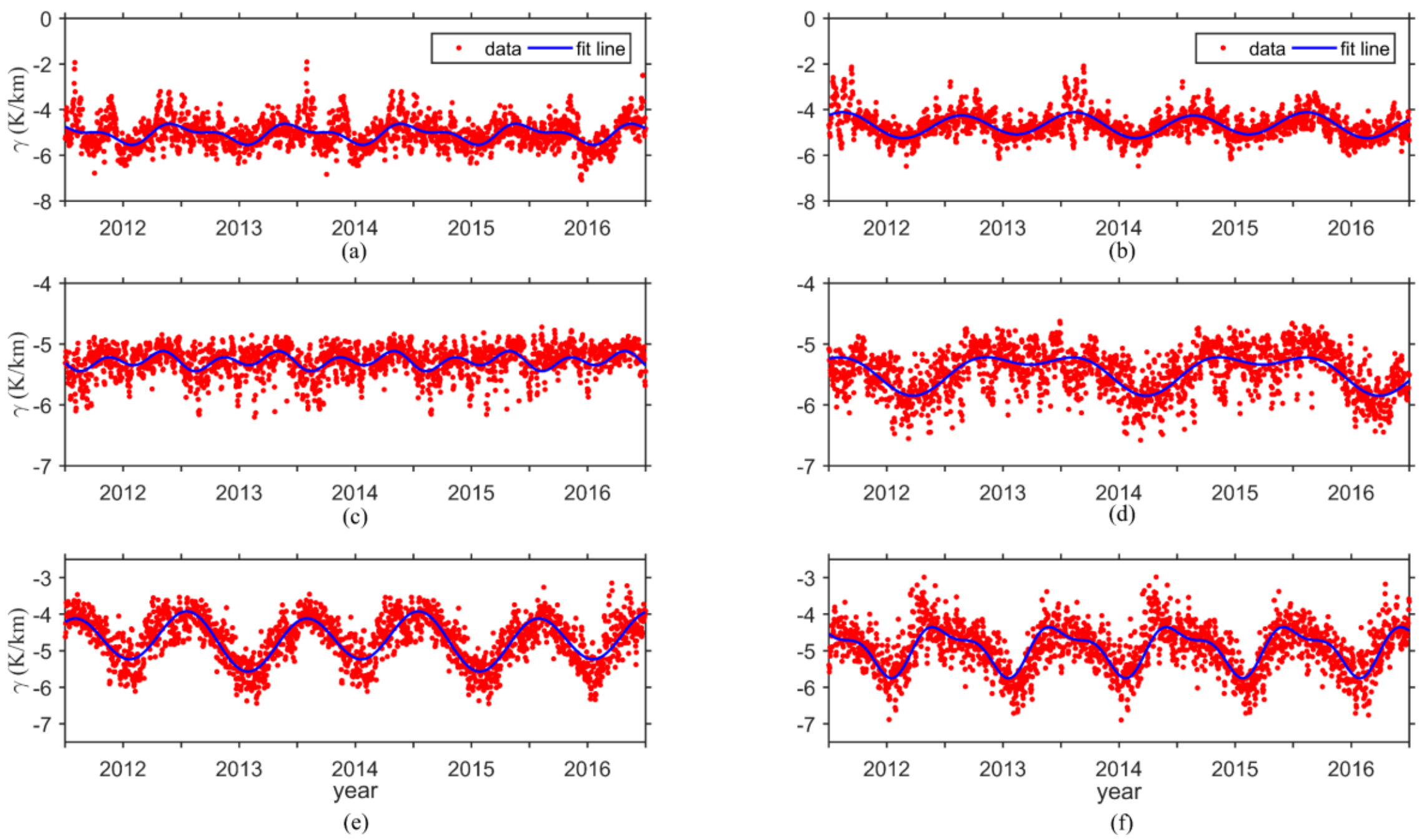

**Figure 2 The time-series variations in the $T_m$ lapse rate from the ERA5 reanalysis data at six representative grid points. (a) (60° N, 90° W). (b) (60° N, 90° E). (c) (0°, 90° W). (d) (0°, 90° E). (e) (60° S, 90° W). (f) (60° S, 90° E).**

Figure 2 shows the obvious seasonal variations in the lapse rate of $T_m$ calculated using the ERA5 reanalysis data at six representative grid points. From Fig. 2, the annual and semiannual variations in the lapse rate of $T_m$ are relatively slight at the grid points located on the equator. However, the lapse rates of $T_m$ at the grid points located in the high-latitude areas of the Southern Hemisphere exhibit relatively large variation ranges and show obvious annual and semiannual variations, whereas those in the high-latitude areas of the Northern Hemisphere show slight variation ranges and obvious annual and semiannual cycle variations. The main reason for these results is that most of the high-latitude areas of the Southern Hemisphere are oceans and are thus not affected by complex climates.

Hence, a clear seasonal pattern is evident in the lapse rate of $T_m$, and the variation patterns vary across different regions. The lapse rate of $T_m$ was then calculated with a temporal resolution of 1 hour from 2012 to 2016 at the global scale. The

annual mean value, annual cycle amplitude, semiannual cycle amplitude and daily cycle amplitude of the lapse rate of $T_m$ were calculated by using Eq. (8) at selected grid points at the global scale. The utilized formula is expressed as follows:

$$\gamma = A_0 + A_1 \cos\left(2\pi \frac{DOY}{365.25}\right) + A_2 \sin\left(2\pi \frac{DOY}{365.25}\right) + A_3 \cos\left(4\pi \frac{DOY}{365.25}\right) + A_4 \sin\left(4\pi \frac{DOY}{365.25}\right) + A_5 \cos\left(2\pi \frac{HOD}{24}\right) + A_6 \sin\left(2\pi \frac{HOD}{24}\right) \quad (8)$$

where $\gamma$ is the lapse rate of $T_m$; $A_0$ is the annual mean value of the lapse rate of $T_m$; ($A_1$, $A_2$) are the annual cycle coefficients for the lapse rate of $T_m$; ($A_3$, $A_4$) are the semiannual cycle coefficients for the lapse rate of $T_m$; ($A_5$, $A_6$) are the daily cycle coefficients for the lapse rate of $T_m$; $DOY$ is the day of the year; and $HOD$ is the UTC time. The above coefficients were calculated at each grid point based on a least-square adjustment by using all selected grid points in the world from 2012 to 2016. The results are shown in Fig. 3.

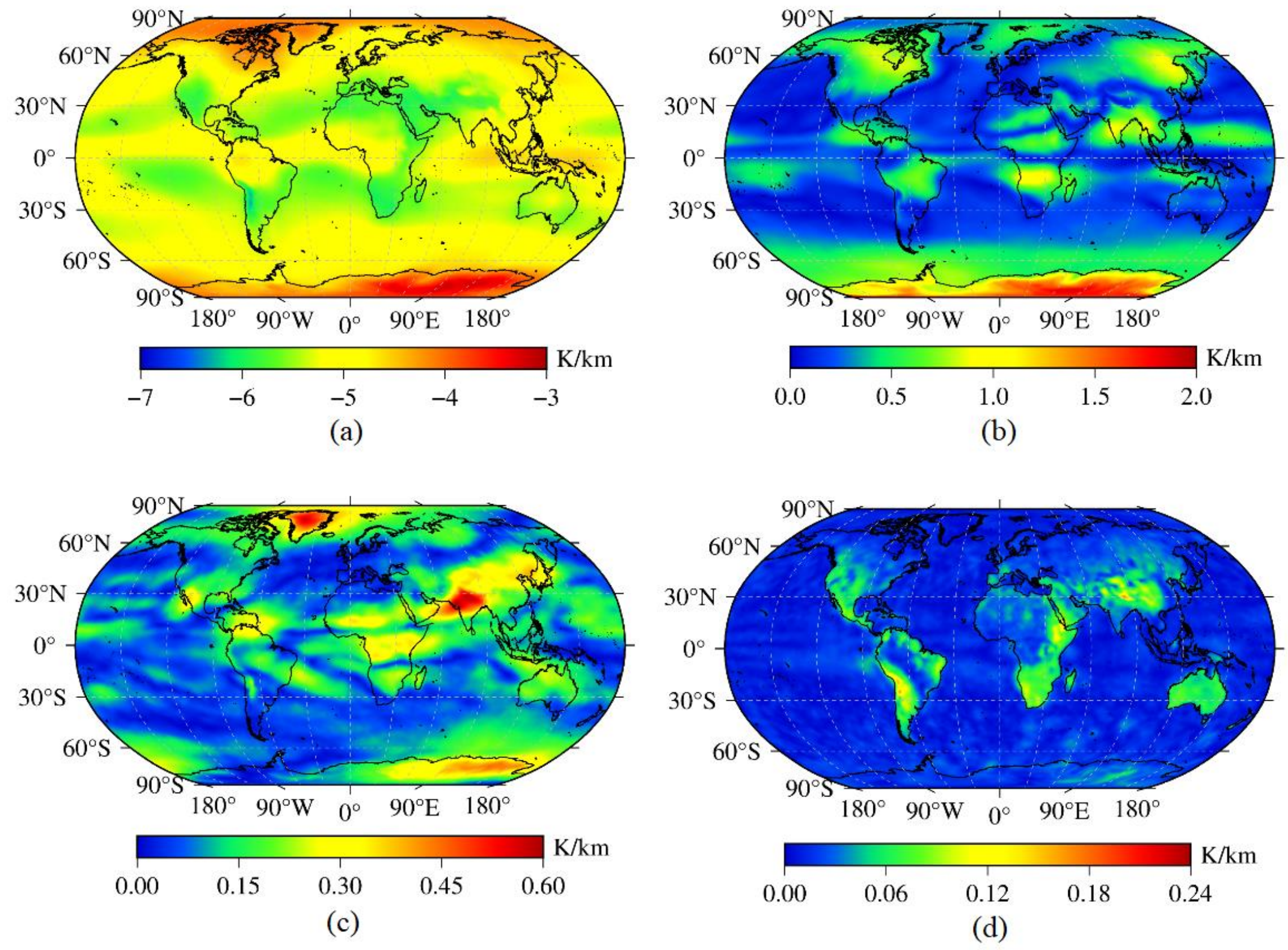

**Figure 3 The distributions of the annual mean value and amplitudes of the $T_m$ lapse rate calculated using global ERA5 reanalysis data. (a) annual average value. (b) annual cycle amplitude. (c) semiannual cycle amplitude. (d) daily cycle amplitude.**

As shown in Fig. 3, a strong correlation was found between the annual mean $T_m$ lapse rate and latitude. Regarding the annual cycle amplitude of the lapse rate of $T_m$, obvious annual cycle amplitude values were observed in most land areas, especially over the Antarctic continent, though these amplitudes were relatively small in the ocean and coastal areas located in the middle and low latitudes of the Northern and Southern Hemispheres. In addition, a sea–land difference was observed in the semiannual cycle amplitude of the $T_m$ lapse rate. The daily cycle amplitude of the lapse rate of $T_m$ remained at approximately 0.06 K/km. Since the daily variation in the lapse rate of $T_m$ can be overshadowed by the annual and semiannual variations, we focused on optimizing the model coefficients solely for these cycles to improve the calculation efficiency when developing the $T_m$ lapse rate model.

The above analysis demonstrated that the variation law of the lapse rate of $T_m$ differs spatially. Moreover, due to the poor performance in spatial differences arising, it is difficult to accurately grasp the variation law of the lapse rate of $T_m$ in developing a global uniform model for the lapse rate of $T_m$. Therefore, we presents a solution to the issue of coefficient redundancy that can occur when developing a model from individual grid points. Specifically, a sliding window algorithm was introduced to develop the $T_m$ lapse rate model, leading to optimized coefficients and improved accuracy, stability, and applicability of the model. Note that, the sliding window algorithm has been used in the previous study, which exhibits a superior performance (Huang et al., 2019a).

### 3.2 Development of NGGTm-H

The ERA5 reanalysis data with a horizontal resolution of 0.25°×0.25° were selected as the data source to develop the model in this study. We divided global segments into regular windows with the same horizontal resolution as the ERA5 reanalysis data. The specific process was as follows: starting from the first window, by using the data of 9 gridded points in each window, the model coefficients of the corresponding window were calculated in order from west to east and from north to south and stored at the geometric center of the corresponding window. Finally, all the coefficients for the global $T_m$ lapse rate model were obtained.

To investigate the influence of the window size on the model precision and optimize the model coefficients as much as possible, three different window sizes, with resolutions of 0.5°×0.5°, 1°×1° and 2°×2°, were selected to develop the model. As mentioned above, it was necessary to consider the characteristics of the annual and semiannual cycles when developing the model. Therefore, the formula of the global $T_m$ lapse rate model in each window can be expressed as follows:

$$\gamma^i = {A_0}^i + {A_1}^i \cos\left(2\pi \frac{DOY}{365.25}\right) + {A_2}^i \sin\left(2\pi \frac{DOY}{365.25}\right) + {A_3}^i \cos\left(4\pi \frac{DOY}{365.25}\right) + {A_4}^i \sin\left(4\pi \frac{DOY}{365.25}\right) \tag{9}$$

where $i$ is the number of windows; $\gamma^i$ is the lapse rate of $T_m$ in the ith window; ${A_0}^i$ is the annual mean value of the lapse rate of $T_m$ in the ith window; (${A_1}^i$, ${A_2}^i$) is the annual cycle coefficient of the lapse rate of $T_m$ in the ith window; (${A_3}^i$, ${A_4}^i$) is the semiannual cycle coefficient of the lapse rate of $T_m$ in the ith window; and $DOY$ is the day of the year.

$$T_m^U = T_m^G - \gamma^i (H^U - H^G) \tag{10}$$

where $T_m^U$ is the $T_m$ value at the user height; $T_m^G$ is the $T_m$ value at the height of the gridded points from the reanalysis data; $\gamma^i$ is the lapse rate of $T_m$ at the window where the user is located; $H^U$ is the elevation of the user; and $H^G$ is the elevation of the gridded point from the reanalysis data.

The five coefficients required in the $T_m$ lapse rate model in all windows of the world were calculated by the least-squares adjustment. Then, the above coefficients were stored at the geometric centers of the windows with resolutions of 0.5°×0.5°, 1°×1° and 2°×2°. Finally, a global real-time and high-precision $T_m$ lapse rate model was developed and named NGGTm-H (this model contains three models with different resolutions: NGGTm-H1, NGGTm-H2 and NGGTm-H3). The vertical $T_m$ correction was calculated by combining Eq. (9) and (10) and using the position and day of the year provided by the users.

### 3.3 Validation of NGGTm-H

To validate the precision and applicability of the spatial interpolation method using the NGGTm-H model at the global scale, the $T_m$ data collected at 378 radiosonde stations around the world in 2017 were used as reference values. The $T_m$ data at four grid points containing radiosonde stations obtained from the surface-level gridded $T_m$ values calculated by the ERA5 reanalysis data were corrected to the heights of the radiosonde stations. Then, the corrected $T_m$ values at these four grid points were interpolated to the positions of the radiosonde stations using the inverse distance-weighted method. Finally, the statistical results of the bias and RMSE values of the spatially interpolated $T_m$ values from all radiosonde stations are shown in Table 1.

**Table 1 The precision statistics obtained for the three resolutions of the NGGTM-H model tested using $T_m$ data from global radiosonde stations and ERA5 surface-level gridded data recorded in 2017 (unit: k)**

| Model | Bias | | | RMSE | | |
|---|---|---|---|---|---|---|
| | Minimum | Maximum | Mean | Minimum | Maximum | Mean |
| NGGTm-H1 | -4.43 | 3.31 | 0.12 | 0.38 | 4.53 | 1.18 |
| NGGTm-H2 | -4.52 | 3.42 | 0.14 | 0.35 | 4.55 | 1.21 |
| NGGTm-H3 | -4.57 | 3.39 | 0.15 | 0.41 | 4.62 | 1.23 |

From Table 1, as the resolution of the model increased, the mean bias of the NGGTm-H model gradually decreased. The mean bias of the NGGTm-H1 model was smallest, at 0.12 K. Compared to those of the NGGTm-H2 model and the NGGTm-H3 model, the mean bias of the NGGTm-H1 model was reduced by only 0.02 K and 0.03 K, respectively. Positive biases with relatively small absolute values were obtained for the NGGTm-H model at the three resolutions using $T_m$ data from radiosonde stations and ERA5 surface-level gridded points. The main reason for these results was that the majority of radiosonde stations are located in land areas. However, the vertical correction values of $T_m$ obtained using the NGGTm-H

model in the land area were slightly larger than the reference values. In addition, the precision of the NGGTm-H1 model showed the best with a mean RMSE of 1.18 K. Thus, NGGTm-H1 model had a certain improvement compared to the NGGTm-H2 and NGGTm-H3 models.

## 4 Development of a global model considering time-varying lapse rate: NGGTm

### 4.1 Analysis of $T_m$ temporal characteristics

Relevant studies have shown that $T_m$ undergoes diurnal variations (Sun et al., 2019). To further analyze the temporal characteristics in $T_m$ in depth at the global scale, we calculated the annual mean value, annual cycle amplitude, semiannual cycle amplitude, daily cycle amplitude, and semidaily cycle amplitude at all grid points using the least-squares adjustment using surface-level gridded $T_m$ data calculated from all the ERA5 reanalysis data recorded from 2012 to 2016 worldwide. The results are shown in Fig. 4.

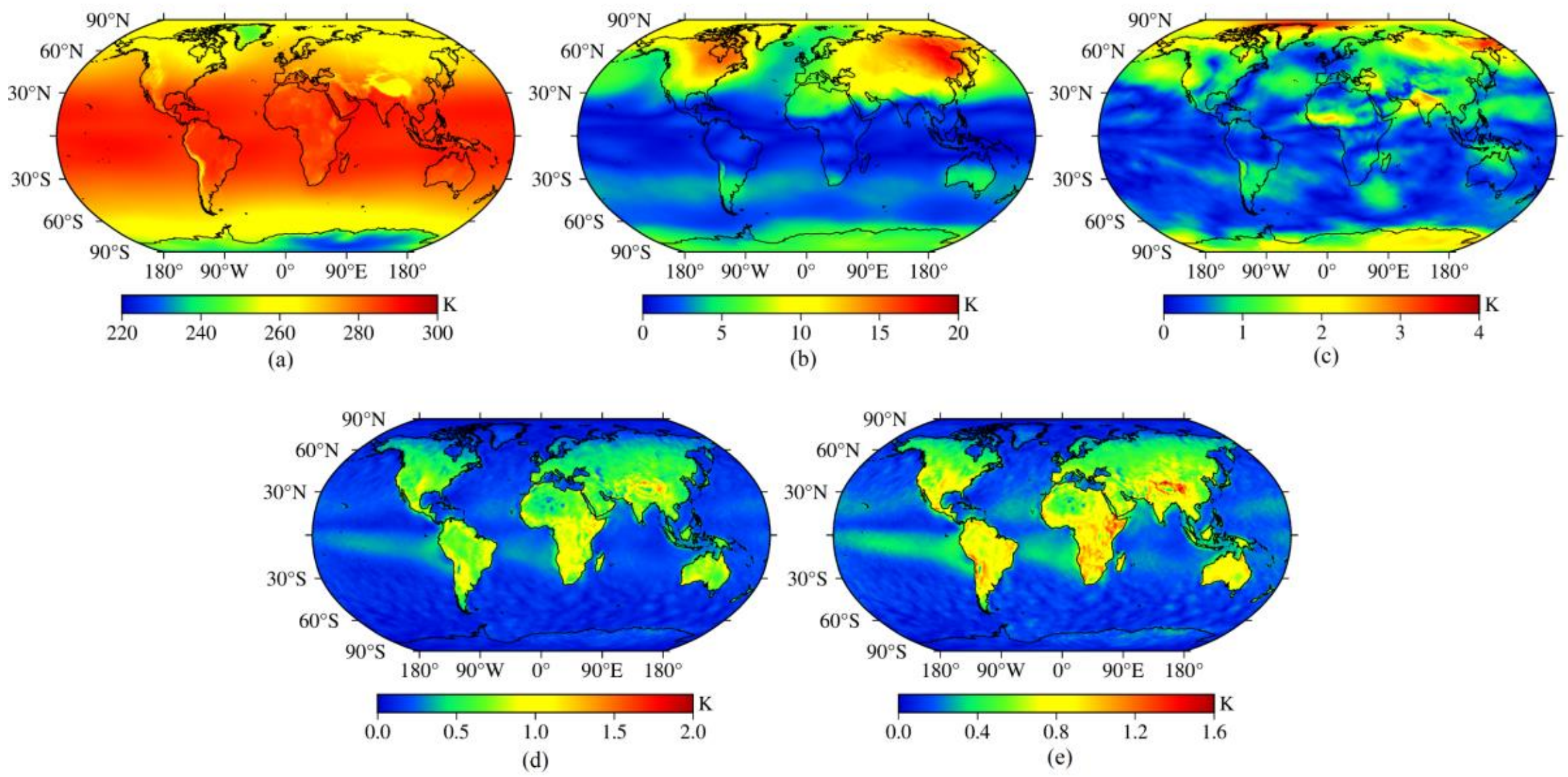

**Figure 4 The distributions of the annual mean value and amplitudes of $T_m$ calculated using global ERA5 reanalysis data. (a) annual mean value. (b) annual cycle amplitude. (c) semiannual cycle amplitude. (d) daily cycle amplitude. (e) half-day cycle amplitude.**

As shown in Fig. 4, strong correlations were found between the annual mean $T_m$ value and latitude and between the annual $T_m$ cycle amplitude and latitude. The semiannual cycle amplitude of $T_m$ also exhibited a certain correlation with latitude, and

a certain sea–land difference was observed. In summary, $T_m$ not only undergoes significant annual and semiannual variations but also experiences significant daily and semidiurnal variation.

### 4.2 Development of the NGGTm model

As mentioned above, it was necessary to consider the time-varying lapse rate and detailed temporal variations when developing high-precision global models. Therefore, a new hybrid-grid global $T_m$ model considering time-varying lapse rate was developed on the basis of NGGTm-H1 model, which used ERA5 reanalysis surface-level data recorded from 2012 to 2016. Since the significant variations in the horizontal direction of $T_m$ compared to lapse rate, the estimation of $T_m$ at the gridded points did not use the sliding window algorithm. The formula is expressed as follows:

$$T_m^G = B_0 + B_1 \cos\left(2\pi \frac{HOD}{24}\right) + B_2 \sin\left(2\pi \frac{HOD}{24}\right) + B_3 \cos\left(4\pi \frac{HOD}{24}\right) + B_4 \sin\left(4\pi \frac{HOD}{24}\right) \quad (11)$$

$$B_i = b_{i0} + b_{i1} \cos\left(2\pi \frac{DOY}{365.25}\right) + b_{i2} \sin\left(2\pi \frac{DOY}{365.25}\right) + b_{i3} \cos\left(4\pi \frac{DOY}{365.25}\right) + b_{i4} \sin\left(4\pi \frac{DOY}{365.25}\right) \quad (12)$$

where $T_m^G$ is the $T_m$ value at the gridded points; $B_i$ is the daily variation coefficient of $T_m$; and $HOD$ is the UTC time. After Eq. (12) was used to expand Eq. (11), $b_{ij}$ (i=0,1,2,3,4 and j=0,1,2,3,4), which represents the 25 coefficient terms of the model, was calculated. $DOY$ is the day of the year.

The 25 model coefficients were calculated by the least-squares adjustment at all global reanalysis data grid points, which used the surface-level gridded $T_m$ data with a temporal resolution of 1 hour. The above coefficients were stored on the grid points with a horizontal resolution of 0.25°×0.25°. Finally, the NGGTm model considering time-varying lapse rate was developed. The input parameters for this model are location and time only, which makes it convenient for users. Then, the $T_m$ values at the positions of the users can be calculated by the inverse distance-weighted method using Eq. (9), (10), (11) and (12).

## 5 Validation of NGGTm

### 5.1 Comparison to gridded $T_m$ data

In this section, to validate the accuracy of the new model, NGGTm model was used to calculate the $T_m$ values at all of the grid points at the global scale, which compared with the Bevis and GPT3 model. surface-level gridded $T_m$ data with a temporal resolution of 1 hour derived from the ERA5 reanalysis data in 2017 were selected as reference values. We defined GPT3 model with two horizontal resolutions of 1°×1° and 5°×5° as GPT3-1 and GPT3-5, respectively, which makes it convenient to describe. The $T_s$ data required by the Bevis model to calculate $T_m$ were derived from the GPT3-1 model. The statistical results are shown in Table 2, Fig. 5 and Fig. 6.

**Table 2 The precision statistics of the bias and RMSE values of the four models tested using global surface-level gridded $T_m$ data from the ERA5 reanalysis product in 2017 (unit: k)**

| Model | Bias | | | RMSE | | |
|---|---|---|---|---|---|---|
| | Minimum | Maximum | Mean | Minimum | Maximum | Mean |
| Bevis | -9.11 | 9.64 | 0.66 | 0.58 | 9.78 | 3.34 |
| GPT3-5 | -15.61 | 22.88 | -0.30 | 0.73 | 23.12 | 3.02 |
| GPT3-1 | -10.13 | 11.51 | -0.28 | 0.68 | 12.88 | 2.90 |
| NGGTm | -1.35 | 1.59 | -0.09 | 0.72 | 6.33 | 2.84 |

From Table 2, it can be seen that the mean bias of the Bevis model was 0.66 K, which indicated that the $T_m$ values calculated by the Bevis model were all larger than the reference values. The mean biases of the GPT3-5 model and the GPT3-1 model were -0.30 K and -0.28 K, respectively, which demonstrated that the $T_m$ values calculated by the GPT3 model were slightly smaller than the reference values. The mean bias of the NGGTm model was only -0.09 K, which was the smallest absolute mean bias value among all the analyzed models. This result shows that the $T_m$ values calculated by this model were close to the reference values overall, which demonstrated that NGGTm model performed better than the other models. In terms of the variation ranges of the bias, the bias variation range of the GPT3-1 model shows improvement compared to that of the GPT3-5 model, which had the largest bias variation range. The main reason for the above results may be the GPT3 model did not consider the influence of elevation in its calculation of $T_m$, which resulted in the relatively large bias in the calculated $T_m$ values in high-elevation areas. The variation range of the bias for the GPT3-1 model was smaller than that of the GPT3-5 model, which indicated that improving the model resolution can help improve the stability of the model. Compared with the Bevis model, GPT3-5 model and GPT3-1 model, the variation range of the bias of the NGGTm model was extremely small, ranging from -1.35 K to 1.59 K, which indicated that the stability of the NGGTm model was better than those of the other analyzed models. In addition, the mean RMSE of the NGGTm model was only 2.84 K, which exhibited improvements of 0.5 K, 0.18 K and 0.06 K over the Bevis model, the GPT3-5 model and the GPT3-1 model, respectively. These results show that the $T_m$ values calculated by the NGGTm model had the highest precision among all analyzed models. In terms of the variation ranges of RMSE, the variation ranges of RMSE for the GPT3-5 model and GPT3-1 model were larger than those of the other models. The RMSE variation range of the GPT3-1 model was smaller than that of the GPT3-5 model. Compared with the other models, the RMSE of the NGGTm model had the smallest variation range, ranging from 0.72 K to 6.33 K, which demonstrated that the precision and stability of the NGGTm model were better than those of the other analyzed models.

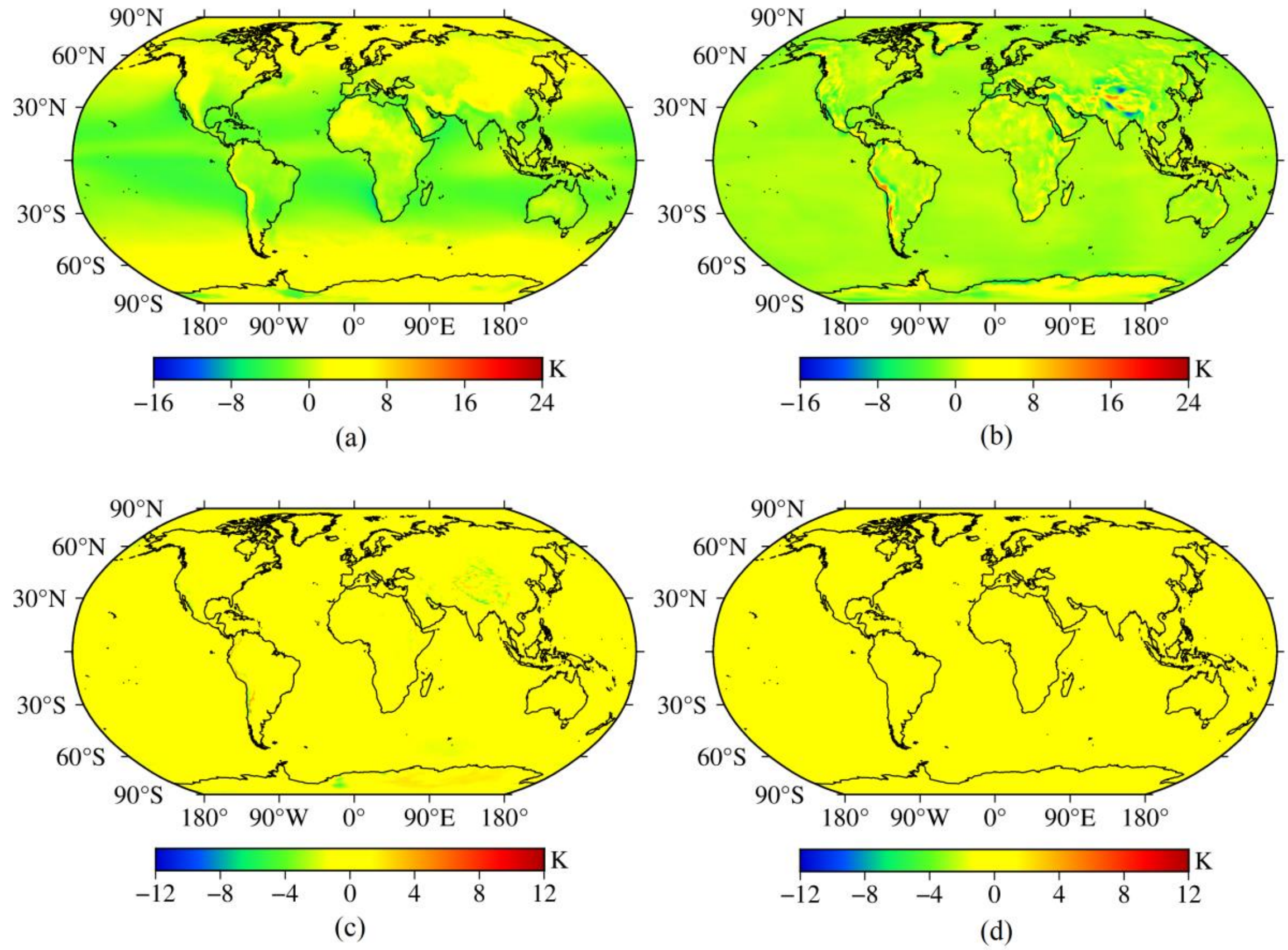

**Figure 5 The bias distributions of the four models tested using global surface-level gridded $T_m$ data from the ERA5 reanalysis product in 2017. (a) Bevis model. (b) GPT3-5 model. (c) GPT3-1 model. (d) NGGTm model.**

Figure 5 shows that the absolute bias values were relatively small for the Bevis model in the mid-latitude areas. The main reason for this result may be the Bevis model was developed based on radiosonde data in North America. Larger absolute bias values were observed for the GPT3-5 model in relatively high-elevation areas, such as the Qinghai-Tibet Plateau, western South America, and parts of Antarctica. The main reason for this result may be the GPT3 model did not take any vertical $T_m$ correction into account. The absolute bias values of the GPT3-1 model were smaller than those of the GPT3-5 model in most parts of the world. Although relatively large absolute bias values were still shown for the GPT3-1 model in relatively high-elevation areas, a significant improvement can be seen. Therefore, the performance and stability of the model can be significantly improved by increasing the resolution of the model. The bias of the NGGTm model remained at approximately 0 K, significantly better than those of other models. In conclusion, the NGGTm model shows excellent stability and applicability at the global scale.

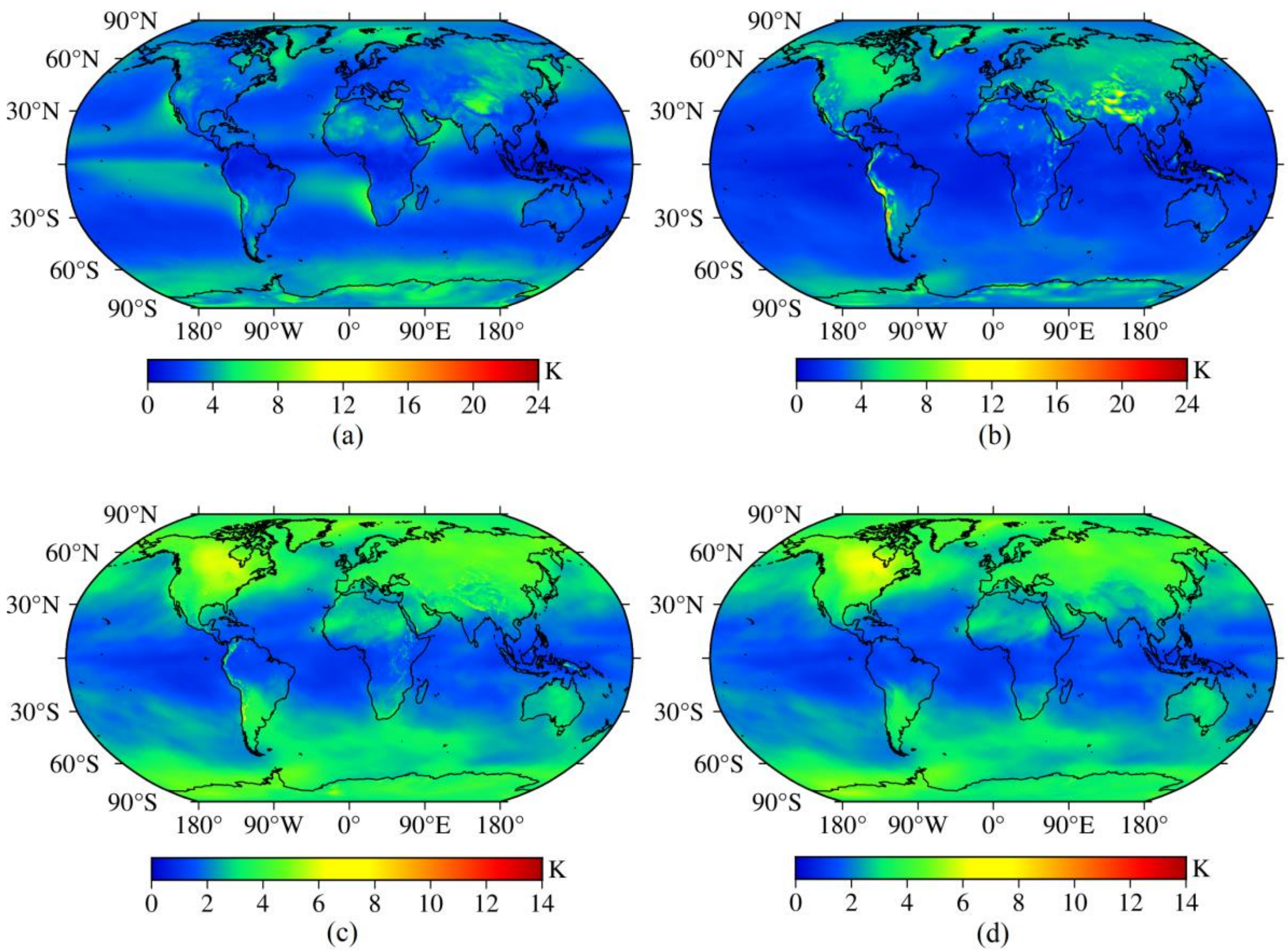

**Figure 6 The RMSE distributions of the four models tested using global surface-level gridded $T_m$ data from the ERA5 reanalysis product in 2017. (a) Bevis model. (b) GPT3-5 model. (c) GPT3-1 model. (d) NGGTm model.**

From Fig. 6, relatively large RMSEs obtained for the Bevis model are shown in some areas, which indicated that the Bevis model performed poorly in the Qinghai-Tibet Plateau, northeastern Asia, the coasts of western and southwestern Africa, the Arctic Ocean, and Antarctica. Relatively large RMSEs obtained for the GPT3-5 model are shown in western North America, western South America, and the Qinghai-Tibet Plateau. The GPT3-1 model had a certain accuracy improvement compared with the GPT3-5 model in these areas. Furthermore, increasing the resolution of the model can improve the model precision of calculating $T_m$. The NGGTm model had small RMSEs around the world, which demonstrated higher precision especially at low latitudes. The NGGTm model performed significantly better than the Bevis model, the GPT3-5 model and GPT3-1 model.

## 5.2 Comparison to radiosonde data

To further validate the performance of the NGGTm model, the $T_m$ data from 378 radiosonde stations around the world in 2017 were selected as reference values. The precision of the NGGTm model when calculating $T_m$ at these stations was validated and compared with the other three models. The $T_s$ data required by the Bevis model to calculate $T_m$ were derived from the radiosonde stations. The statistical results are shown in Table 3, Fig. 7 and Fig. 8.

**Table 3 The precision statistics of bias and RMSE for the four models tested using global $T_m$ data from 378 radiosonde stations in 2017 (unit: k)**

| Model | Bias | | | RMSE | | |
|---|---|---|---|---|---|---|
| | Minimum | Maximum | Mean | Minimum | Maximum | Mean |
| Bevis | -4.98 | 6.49 | 0.39 | 0.98 | 7.05 | 3.57 |
| GPT3-5 | -13.79 | 4.48 | -1.00 | 0.99 | 13.90 | 3.65 |
| GPT3-1 | -5.66 | 4.49 | -0.79 | 0.98 | 6.23 | 3.48 |
| NGGTm | -4.31 | 3.78 | 0.10 | 0.99 | 5.17 | 3.30 |

From Table 3, the mean bias of the NGGTm model was only 0.10 K, with the smallest absolute value among the analyzed models. The bias range of the NGGTm model was also the smallest, ranging from -4.31 K to 3.78 K, which demonstrated that the NGGTm model performed better than the other models. In addition, the mean RMSE of the NGGTm model was only 3.30 K, which exhibited improvements of 0.27 K (8%), 0.35 K (11%) and 0.18 K (5%) over the Bevis model, GPT3-5 model and GPT3-1 model, respectively. The RMSE range of the NGGTm model was the smallest, ranging from 0.99 K to 5.17 K, indicating that the NGGTm model had the best precision and stability at the global scale.

From Fig. 7, the Bevis model showed relatively obvious negative biases in low latitudes and obvious positive biases in middle and high latitudes, with a trend of increasing absolute biases from low latitudes to high latitudes. The GPT3-5 model and GPT3-1 model performed similarly, with relatively large absolute bias values on the Qinghai-Tibet Plateau and in western North America because the GPT3 model did not consider the relationship between $T_m$ and elevation. The absolute bias values of the NGGTm model were relatively small at the global scale, with values of approximately 0 K. These results demonstrated that the stability of the NGGTm model was better than those of the other analyzed models at the global scale.

Figure 8 shows all models exhibited relatively small RMSEs at low latitudes and relatively large RMSE values at high latitudes, with a trend of increasing RMSEs from low latitudes to high latitudes. The main reason for this result may be the seasonal variation in $T_m$ is strengthened with increasing latitude. In addition, the GPT3-5 model showed relatively large RMSEs at a few radiosonde stations on the Qinghai-Tibet Plateau, whereas the GPT3-1 model exhibited a certain improvement for the reasons mentioned above. The NGGTm model still had a significant improvement compared with other models at the global scale, which demonstrated that the NGGTm model had the best precision and stability.

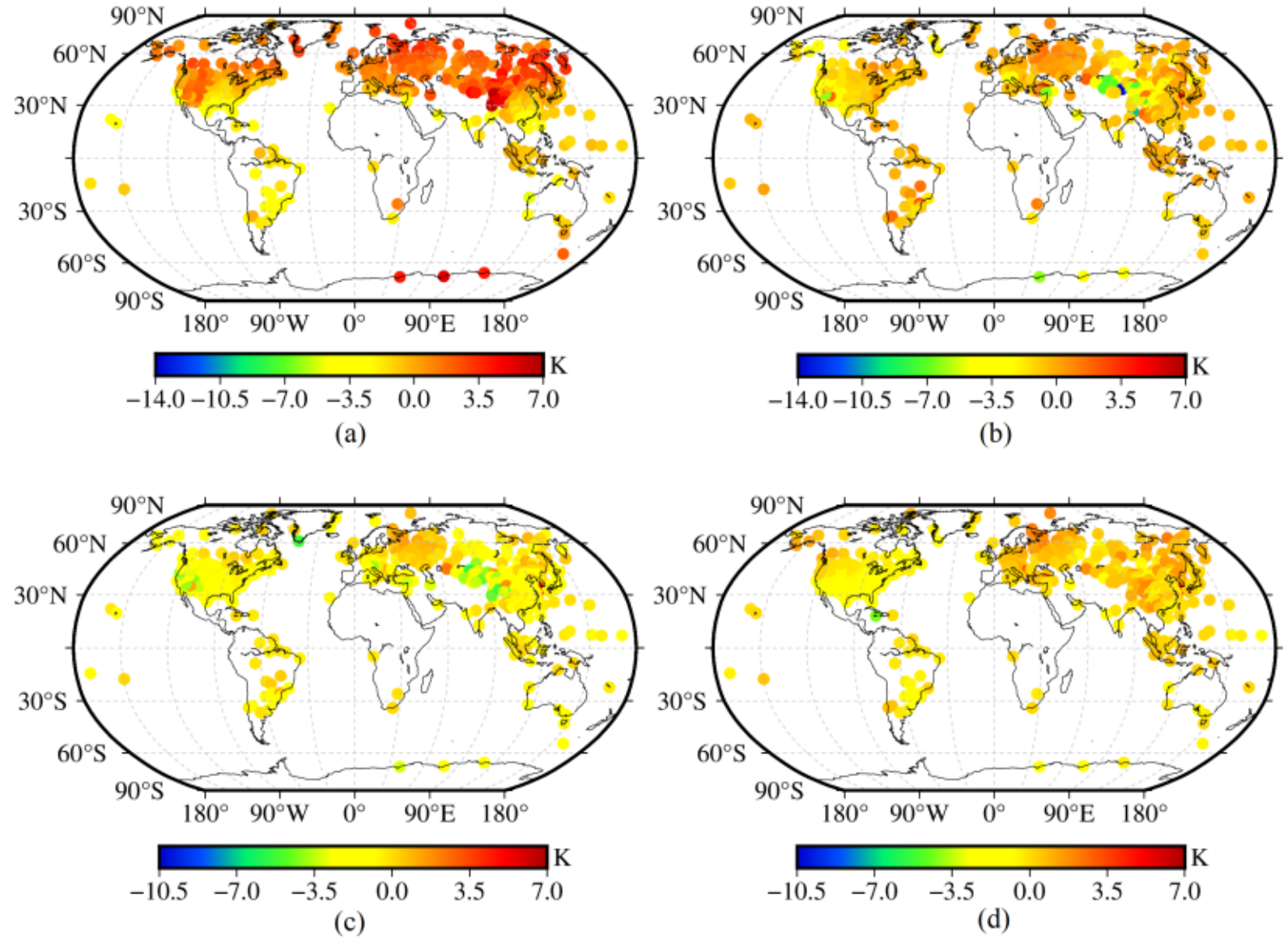

**Figure 7 The bias distributions of the four models tested using global $T_m$ data from 378 radiosonde stations in 2017. (a) Bevis model. (b) GPT3-5 model. (c) GPT3-1 model. (d) NGGTm model.**

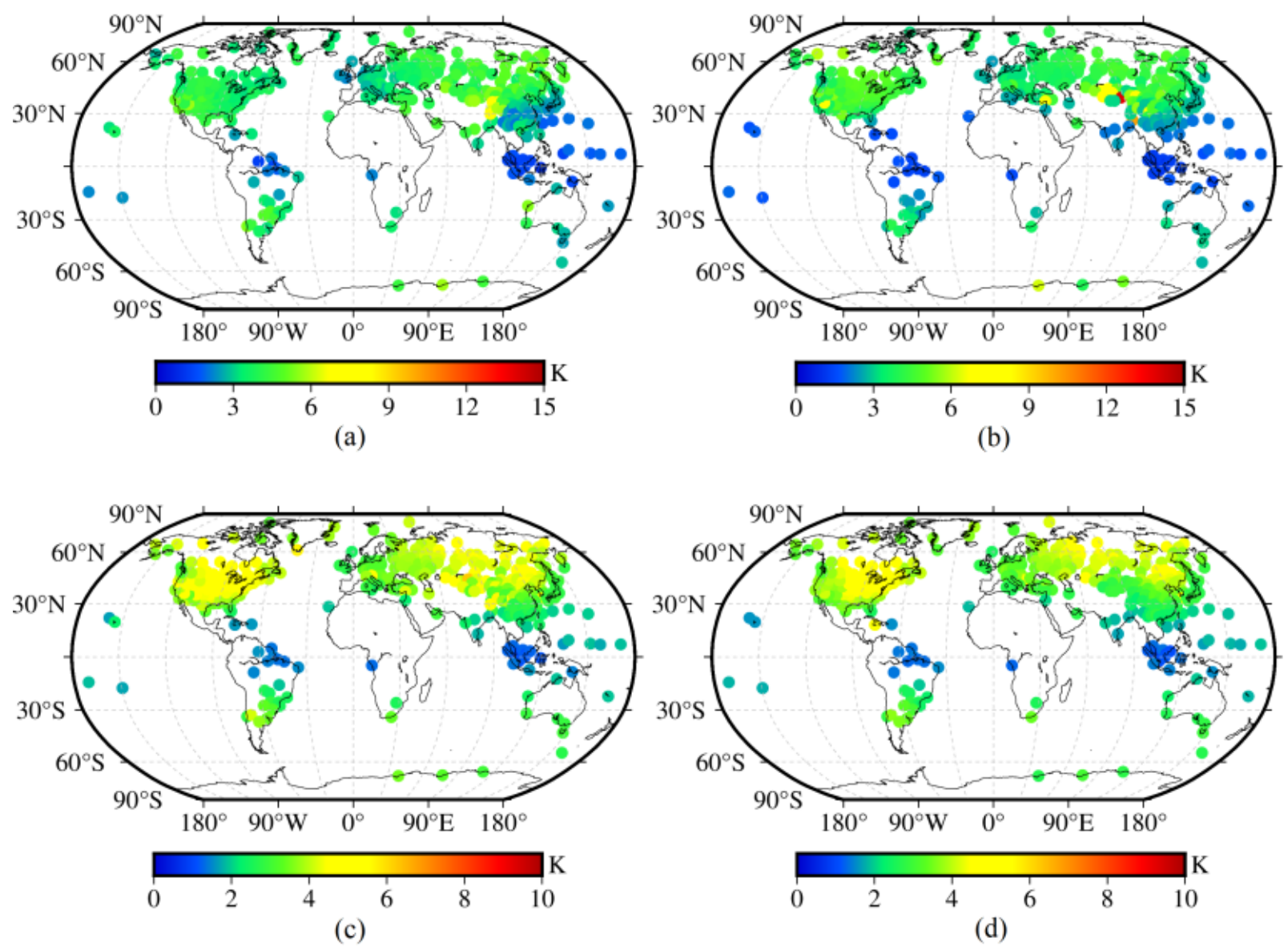

**Figure 8 The RMSE distributions of the four models tested using global $T_m$ data from 378 radiosonde stations in 2017. (a) Bevis model. (b) GPT3-5 model. (c) GPT3-1 model. (d) NGGTm model.**

Since there are strong correlations between $T_m$ and both elevation and latitude, to further analyze the relationship between the precision of $T_m$ calculated by the four models and the elevation variation, the 378 radiosonde stations around the world were divided into five intervals with an elevation span of 500 m for each interval. The bias and RMSE values at these 378 radiosonde stations around the world were then calculated according to the above intervals. The results are shown in Fig. 9.

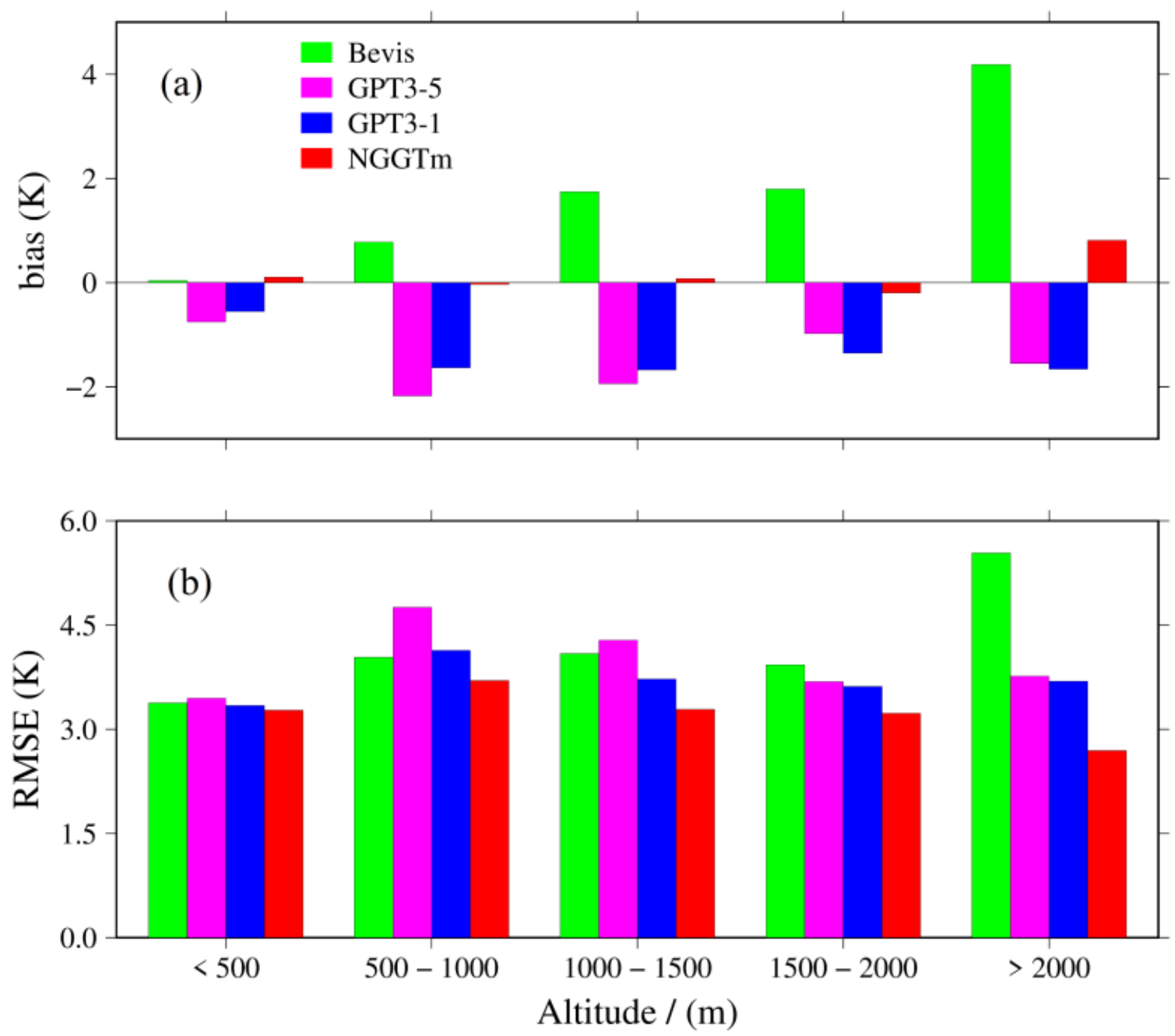

**Figure 9 The bias and RMSE distributions within different elevation intervals for the four models tested using global $T_m$ data from 378 radiosonde stations in 2017. (a) bias. (b) RMSE.**

From Fig. 9, the Bevis model exhibited a positive correlation between bias and elevation. The main reason for the above result may be the Bevis model was developed by using radiosonde data collected in low-elevation North America, which leading the poor applicability in relatively high-elevation areas. The GPT3-5 model and GPT3-1 model exhibited negative biases in all elevation intervals. Whereas the NGGTm model exhibited relatively small absolute biases in all elevation intervals, especially those below 2000 m. Therefore, the NGGTm model exhibited extremely significant stability in all elevation intervals compared with other models at the global scale. In addition, the Bevis model performed smaller RMSEs than the GPT3-5 model at elevations below 1500 m. The RMSEs of the GPT3-1 model were smaller than those of the GPT3-5 model at all elevation intervals, which further indicatied that increasing the resolution of the model can improve the precision and stability of the results. The RMSEs of the NGGTm model were smaller than those of the Bevis model, GPT3-5

model and GPT3-1 model in all elevation intervals. In conclusion, the NGGTm model showed the best precision and stability compared with the other analyzed models in all elevation intervals.

To further analyze the relationship between the precision of four models and the latitude variation , the 378 radiosonde stations around the world were divided into several intervals with a latitude interval of 15 degrees. Few radiosonde stations are located at high latitudes. The high-latitude areas in the Northern and Southern Hemispheres were divided into intervals with latitude intervals of 15 degrees. The bias and RMSE values corresponding to the 378 radiosonde stations around the world were calculated according to the above intervals. The results are shown in Fig. 10.

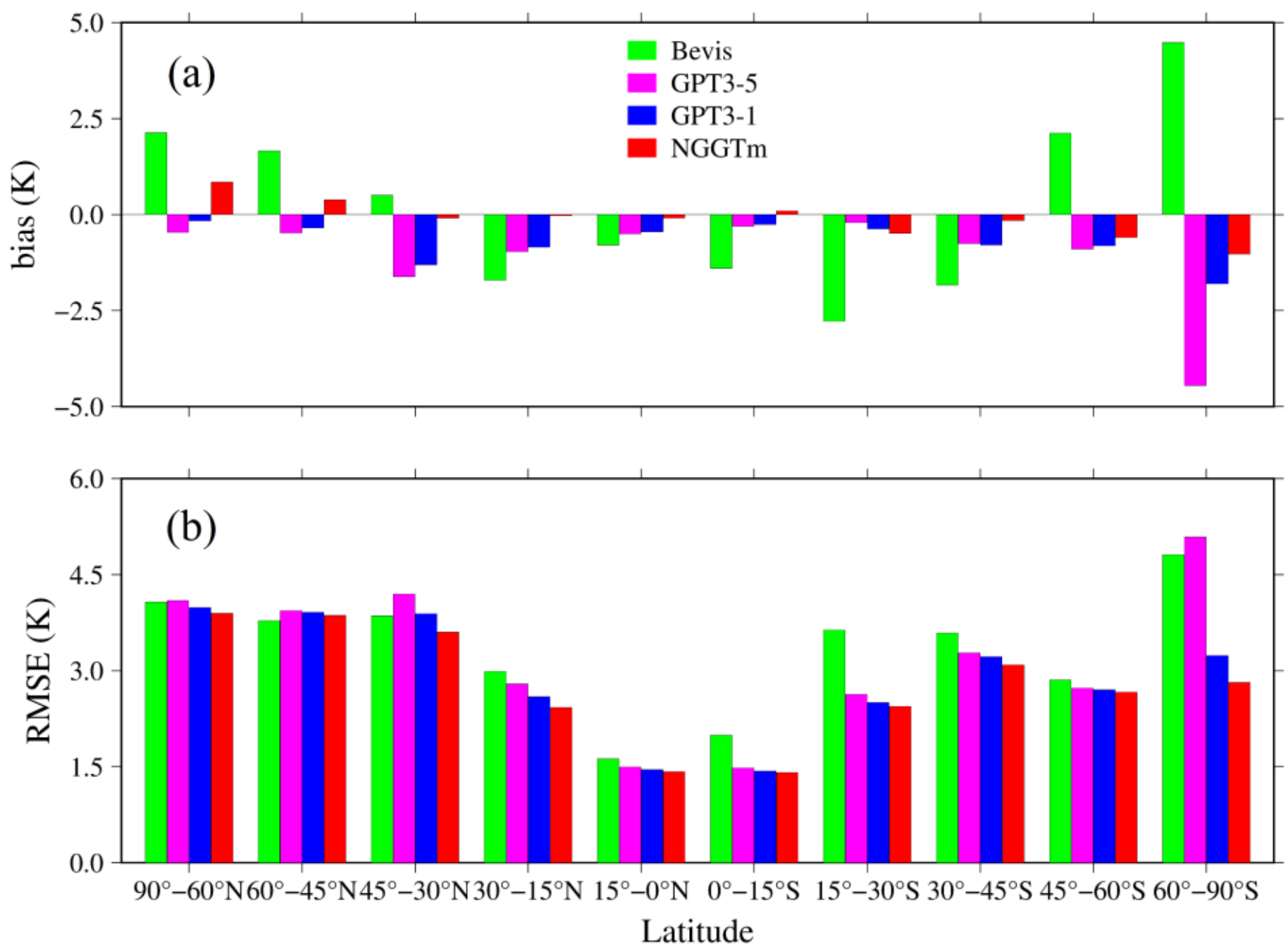

**Figure 10 The bias and RMSE distributions in different latitude ranges obtained for the four models tested using $T_m$ data recorded at 378 radiosonde stations globally in 2017. (a) bias. (b) RMSE.**

From Fig. 10, the Bevis model obtained relatively large absolute biases in most latitude ranges, which exhibited significantly positive biases in high latitudes and significantly negative biases in low latitudes. The GPT3-5 and GPT3-1 models exhibited negative biases with relatively small absolute values at most latitudes and negative biases with relatively large absolute values in the high-latitude areas of the Southern Hemisphere, which can be observed especially for the GPT3-5 model. The NGGTm model exhibited small absolute biases in all latitude ranges. In addition, all models showed small RMSEs in low-latitude areas but relatively large RMSEs in high-latitude areas. The RMSEs gradually increased with increasing latitude for all models. Compared to the other models, the NGGTm model had the best accuracy in all latitude
ranges, especially in the high-latitude areas of the Southern Hemisphere. In summary, the NGGTm model showed a high accuracy and stability for calculating $T_m$ at all latitudes.

**6 Conclusion**

$T_m$ is the key parameter of GNSS PWV inversion tasks and in the detection of PWV changes. Developing a real-time and high-precision $T_m$ lapse rate model is necessary for $T_m$ vertical correction. By analyzing the relationship between $T_m$ and elevation, an approximately linear relationship between $T_m$ and elevation can be found in the near-Earth space. Therefore, a linear function was used to fit the lapse rate of $T_m$. Based on an in-depth analysis of the detailed temporal variations in the $T_m$ lapse rate, a sliding-window algorithm was introduced to develop the NGGTm-H model with horizontal resolutions of 0.5°×0.5°, 1°×1° and 2°×2°. The user can obtain the corresponding vertically corrected $T_m$ value by providing only the coordinate information of any position and the day of the year. The NGGTm-H model can achieved excellent results in the precision verification performed by combining ERA5 reanalysis data and radiosonde data that were not involved in the modeling process.

Based on the development of the $T_m$ lapse rate model and taking into account the impacts of the detailed temporal characteristics of $T_m$, NGGTm model was developed. The accuracy and applicability of the NGGTm model were then verified by global radiosonde data and ERA5 reanalysis data that were not involved in the modeling process, which compared with those of the Bevis model and GPT3 model. The results show that the NGGTm model had the best performance and stability among the tested models. Compared to the Bevis model and GPT3 model, with increasing elevation, the performance improvement of the NGGTm model was more significant. The accuracy of the NGGTm model was also significantly improved with increasing latitude. In general, the NGGTm model can provide real-time and high-precision $T_m$ information without requiring the input of measured meteorological parameters at the global scale. This model has broad application prospects in real-time GNSS PWV detection research.

This study only verifies the stability and applicability of the NGGTm model, whereas the model has not yet been applied to GNSS PWV retrieval tasks. Therefore, the effectiveness of the NGGTm model in retrieving atmospheric PWV will be further investigated in future study.

*Data availability.* The ERA5 reanalysis data used in this paper can be freely accessed at (http://cds.climate.copernicus.eu/cds app#!/search?text=ERA5). The radiosonde data can be accessed at (http://weather.uwyo.edu/upperair/sounding.html).

*Author Contributions.* Shaofeng Xie: Conceptualization, Methodology, Formal analysis, Validation, Data curation, Writing-original draft, Writing-review & editing, Funding acquisition. Jihong Zhang: Conceptualization, Methodology, Formal analysis, Software, Validation, Data curation, Writing-original draft. Liangke Huang: onceptualization, Methodology,

Formal analysis, Data curation, Writing-review & editing. Fade Chen: Validation. Yongfeng Wu: Investigation. Yijie Wang: Investigation. Lilong Liu: Investigation, Funding acquisition.

*Competing interests.* The contact author has declared that none of the authors has any competing interests.

*Acknowledgments.* The authors would like to thank to the European Centre for Medium-Range Weather Forecasts (ECMWF) for providing ERA5 reanalysis data and the University of Wyoming for providing radiosonde data.

*Financial support.* This work was supported by the Guangxi Natural Science Foundation of China (2023GXNSFAA026434).

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
