# Peer review of "A hybrid-grid global model for the estimation of atmospheric weighted mean temperature considering time-varying vertical adjustment rate in GNSS precipitable water vapor retrieval"

_Geoscientific Model Development, 2024_

## Author Response (AR1)

**Response to Editor and Reviewers**

Dear editor and reviewers,

Thank you for offering us an opportunity to improve the quality of our submitted manuscript titled "**A hybrid-grid global model for the estimation of atmospheric weighted mean temperature considering time-varying lapse rate in GNSS precipitable water vapor retrieval**" (gmd-2024-21). We appreciated very much the reviewers' constructive and insightful comments. In the following, we include a point-by-point response to the comments from each reviewer. In the revised manuscript, all the changes have been highlighted in red. We hope the revised manuscript has now met the publication standard of your journal.

**Reviewer #1**

**Comment 1:** Please correct the grammar errors, typo, or missing words in the For example, in line 109, "we aim to global Tm model that takes into account..." and in line 298, "surface-level…".

**Response 1:** Thank you for pointing this out. We agree with this comment. We have corrected the grammar errors, typo, and missing words. In the revised manuscript, the relevant content has been modified as follows: "our aim was to develop a global Tm model that takes into account time-varying lapse rate and high-precision capabilities" (see lines 108-109), "The surface gridded Tm data with a temporal resolution of 1 hour derived from the ERA5 reanalysis data in 2017 were selected as reference values" (see lines 312-313). In addition, we have carefully read the entire manuscript and corrected the remaining errors.

**Comment 2:** In section 3.3, the NGGTm-H model is validated by radiosonde. I suggest describing the vertical resolutions or the altitude of the record at the 378 radiosonde stations.

**Response 2:** Thanks for your suggestion. In Section 3.3, we have added descriptions of the altitude of the record at the 378 radiosonde stations. In the revised manuscript, the relevant content has been modified as follows: "The altitude of radiosonde stations ranges from 0 to 4500 m, mostly within 2000 m" (see lines 249-250).

**Comment 3:** In line 256-260, do the authors try to explain the reason why the positive biases are smaller than the absolute value of the negative biases? If yes, please explain the reason in more detail.

**Response 3:** Thanks for your comment. We have tried to explain the reason why the positive biases are smaller than the absolute value of the negative biases. The relevant content has been added in the revised manuscript: "The vertical correction values of Tm obtained using the

NGGTm-H model were slightly larger in land areas but smaller in marine areas than the reference values. However, a small number of radiosonde stations distributed in marine areas were susceptible to the influence of marine climate, resulting in the vertical correction values of the model was apparently smaller than the reference values. Therefore, the positive biases were smaller than the absolute value of the negative biases" (see lines 262-266). In future work, we will further investigate the reasons and develop solutions.

**Comment 4:** I suggest using the same color bar ranges in (a)-(d) of figure 5 and 6 to emphasize the result and avoid misunderstanding to readers, especially in figure 6.

**Response 4:** Thank you for pointing this out. We have changed the color bar ranges in two figures to be consistent (see lines 340 and 353). Indeed, it will mislead readers if the color bar ranges are inconsistent.

**Reviewer #2**

**Comment 1:** L15, What does "NGGTm-H" stand for?

**Response 1:** Thanks for the question you raised. The "NGGTm-H" stands for a new global grid Tm lapse rate model. Lapse rate is the rate at which Tm decreases with increasing height. We have added explanations in the revised manuscript (see lines 14, 16 and 110).

**Comment 2:** L39-40, "Microwave radiometers and satellite remote sensing, which rely on infrared band detection, offer high detection accuracies." This is logically misleading. Microwave radiometers operate in the microwave region of the electromagnetic spectrum, not the infrared.

**Response 2:** Thank you for pointing this out. We agree with this comment. In the revised manuscript, the relevant content has been modified as follows: "Microwave radiometers that operate in the microwave region of the electromagnetic spectrum, and satellite remote sensing that rely on infrared band detection, offer high detection accuracies" (see lines 38-40).

**Comment 3:** L45, "a high spatiotemporal resolution" this is not completely right. Globally speaking, the number of GNSS stations is still low.

**Response 3:** Thank you for pointing this out. We agree with this comment. We have replaced "a high spatiotemporal resolution" with "a high temporal resolution" (see line 44).

**Comment 4:** L54, "However, the variation law of ZWD influenced mainly by water…" What is the variation law of ZWD? If there is such a law, then the ZWD variation can be well known.

**Response 4:** Thanks for your comment. We apologize for that our description is misleading. The meaning we want to express is that the variation in ZWD is mainly influenced by precipitable water vapor (PWV). There is a relationship between ZWD and PWV (ZWD=PWV/K). Variation

in PWV can cause variation in ZWD. In reality, PWV changes rapidly, which leads to rapid changes in ZWD. Therefore, it is difficult to investigate the variation law of ZWD. We do not emphasize which variation law ZWD has. In the revised manuscript, the relevant content has been modified as follows: "However, the variation in ZWD influenced mainly by water vapor is difficult to investigate" (see line 53).

**Comment 5:** L58, "The accuracy of GNSS tropospheric water vapor retrievals can be significantly improved by using high-precision $Tm$ data." How is the accuracy of PWV affected if different Tm values are used? Please provide numerical examples.

**Response 5:** Thanks for the question you raised. Huang et al. (2019) studied the impact of Tm on GNSS-PWV using the relationship of RMSE between Tm and PWV. The results indicated that the RMSE of Tm for their proposed GGTm model was 3.54 K, and the RMSE of inverted PWV was 0.26 mm. However, the RMSE of Tm for Bevis model was 4.1K, and the RMSE of inverted PWV was 0.31 mm.

Huang, L. K., Jiang, W. P., Liu, L. L., Chen, H., and Ye, S. R.: A new global grid model for the determination of atmospheric weighted mean temperature in GPS precipitable water vapor, J. Geod., 93, 159–176, https://doi.org/10.1007/s00190-018-1148-9, 2019.

**Comment 6:** L63, "it is necessary to build a real-time and high-precision $Tm$." I concur that a high-precision Tm is needed. Is it really necessary to build a real-time Tm model?

**Response 6:** Thanks for the question you raised. It is necessary to build a real-time Tm model in the application of GNSS-PWV inversion. PWV is closely related to atmospheric circulation, climate change and extreme rainstorm. Corresponding measures can be taken to address the aforementioned natural phenomena according to the prediction of future PWV content and variations. The Tm models built earlier were meteorological parameter models that relied on measured meteorological parameters, which could not calculate Tm in real-time. Later, many scholars built nonmeteorological parameter models to achieve real-time calculation of Tm and real-time inversion of PWV.

**Comment 7:** L64, "Existing $Tm$ models can be divided into two categories: meteorological parameter models and nonmeteorological parameter models." What are the representative models for the meteorological parameter models and what are the representative models for the nonmeteorological parameter models? Do you have reference papers?

**Response 7:** Thanks for your comment. We listed some representative meteorological and nonmeteorological parameter models after this sentence (see lines 63-88). Representative meteorological parameter models such as Bevis (Bevis et al., 1992) model and GTm-I (Yao et al., 2014c). Representative nonmeteorological parameter models, such as the Emardson model (Emardson & Derks, 2000).

Bevis, M., Businger, S., Herring, T. A., Rocken, C., Anthes, R. A., and Ware, R. H.: Remote sensing of atmospheric water vapor using the Global Positioning System, J. Geophys. Res.: Atmos., 97, 787–15, https://doi.org/10.1029/92JD01517, 1992.

Emardson, T. R. and Derks, H. J.: On the relation between the wet delay and the integrated precipitable water vapour in the European atmosphere, Meteorol. Appl., 7, 61–68, https://doi.org/10.1017/S1350482700001377, 2000.

Yao, Y. B., Zhang, B., Xu, C. Q., and Yan, F.: Improved one/multi-parameter models that consider seasonal and geographic variations for estimating weighted mean temperature in ground-based GPS meteorology, J. Geod., 88, 273–282, https://doi.org/10.1007/s00190-013-0684-6, 2014.

**Comment 8:** L85-89, "The nonmeteorological parameter $Tm$ model (named the Emardson model) was developed to take the annual cycle variation into account by using radiosonde data collected in Europe over many years, which was capable of meeting the requirement for GNSS water vapor detection (Emardson & Derks, 2000). Therefore, the model has been widely used in real-time GNSS meteorology research." The Tm model was developed based on the radiosonde data collected in Europe. Does it really meet the accuracy requirement for GNSS water vapor detection out of Europe?

**Response 8:** Thanks for the question you raised. Based on our experience, the Tm model developed using data from a certain region has higher accuracy in GNSS water vapor detection in this region, but lower accuracy in other regions. To achieve high accuracy in other regions, we can use the data from the target region and refer to the Emardson model expression to develop a new model. Therefore, when the natural geographical conditions of the target regions differ significantly from Europe, the Emardson model may not meet the accuracy requirements of GNSS water vapor detection. In this case, we can consider developing a new Emardson model.

**Comment 9:** L98-100, "This new model can estimate the $Tm$ value at any location by simply inputting the station location and the annual product day, which have been applied to real-time GNSS PWV inversion studies worldwide." Are there any reference papers? What is the accuracy of Tm used in GNSS PWV inversion?

**Response 9:** Thanks for the question you raised. The new model and its applications we described are refered to the article by Yao et al. (2012). They developed the GWMT model and applied it to GNSS PWV inversion. We have not yet found any scholars who have applied the Tm model to GNSS PWV inversion. The Tm model (GWMT model) proposed by Yao et al. (2012) has an internal accuracy of 4 K and an external accuracy of 4.6 K.

Yao, Y. B., Zhu, S., and Yue, S. Q.: A globally applicable, season-specific model for estimating the weighted mean temperature of the atmosphere, J. Geod., 86, 1125–1135, https://doi.org/10.1007/s00190-012-0568-1, 2012.

**Comment 10:** L104-106, "Although the GPT3 model is currently the most representative empirical model with a high precision on the global scale, GPT3 model dose not take into account elevation correction or detailed $Tm$ lapse rate." The current GPT3 can get a high precision on global scale, though it didn't consider the Tm lapse rate. Justify why you need to do this work.

**Response 10:** Thanks for your comment. We compared the accuracy of the proposed NGGTm model (considering the Tm lapse rate) with the GPT3 model (no considering the Tm lapse rate). In Section 5.1, when using gridded data as reference values, the mean RMSE of GPT3-1 and GPT3-5 were 2.90 and 3.02 K, respectively, whereas the mean RMSE of the NGGTm model was 2.84 K. In Section 5.2, when using radiosonde data as reference values, the mean RMSE of GPT3-1 and GPT3-5 were 3.48 and 3.65 K, respectively, whereas the mean RMSE of the NGGTm model was 3.30 K. The accuracy of the NGGTm model is higher than that of the GPT3 model. Therefore, it is necessary to do this work.

**Comment 11:** L109, "we aim to global $Tm$ model that takes into account time-varying…" This sentence is grammatically erroneous.

**Response 11:** Thank you for pointing this out. We agree with this comment. we have corrected the grammar errors. In the revised manuscript, the relevant content has been modified as follows: "our aim was to develop a global Tm model that takes into account time-varying lapse rate and high-precision capabilities" (see lines 108-109).

**Comment 12:** Eqs. (5) and (6), why is the variable KP used? What is the rational of using KP, not Tm?

**Response 12:** Thanks for the question you raised. The meaning of variable KP is key parameter . In order to make it easy for readers to understand its meaning, we have modified it to Tm according to your suggestion (see lines 153-156).

**Comment 13:** The paragraph (around L185) discussed the result of Fig. 2. However it is very hard to understand. You should cite the Fig. 2(a)… Fig. 2(f) in the discussion.

**Response 13:** Thanks for your valuable suggestion. Indeed, this makes it difficult for readers to understand. We have cited specific figures in the discussion (see lines 183-187).

**Comment 14:** L211, "we focused on optimizing the model coefficients solely for these cycles to improve the calculation efficiency when developing the $Tm$ lapse rate model." Why is the calculation efficiency so critical? What is the normal calculation efficiency? Is the current calculation not fast or efficient enough?

**Response 14:** Thanks for the question you raised. After our testing, it takes 650 seconds to calculate the lapse rate of Tm at about 260,000 center points of window and 8760 hours of one year when taking into account annual and semiannual cycles. It takes 630 seconds when considering annual, semiannual, daily, and semidaily cycles. The difference in computational efficiency between the two methods is not significant. Therefore, we believe that there is no need to emphasize computational efficiency here. What we should pay more attention to is the simplification of the model. As mentioned in lines 209-211, the daily variation in lapse rate of Tm can be overshadowed by annual and semiannual variations, so there is no need to consider daily variations. In the revised manuscript, the relevant content has been modified as follows: "Since the daily variation in the lapse rate of Tm can be overshadowed by the annual and semiannual

variations, we focused on optimizing and simplifying the model coefficients when developing the Tm lapse rate model" (see lines 209-211).

**Comment 15:** L219, "Note that, the sliding window algorithm has been used in the previous study, which exhibits a superior performance" What is the difference between the sliding window algorithm in the previous study and that in this submission?

**Response 15:** Thanks for the question you raised. In the previous study, the horizontal resolution of grid data is 2.5°×2° (lon.×lat.) and the sliding window size is 5°×4° (lon.×lat.). In this study, the horizontal resolution of grid data is 0.25°×0.25° (lon.×lat.). To investigate the influence of the window size on the model precision and optimize the model coefficients as much as possible, three different window sizes with resolutions of 0.5°×0.5°, 1°×1° and 2°×2°, were selected to develop the model. Due to the improved horizontal resolution of the grid data used in this study, the size of the sliding window had been adjusted.

**Comment 16:** L224, "by using the data of 9 gridded points in each window," Explain what the 9 gridded points in each window? What is the window? It is better to have figure illustration.

**Response 16:** Thanks for your comment, it is a very valuable suggestion. We have added a figure illustration and explained what 9 gridded points and windows are (see lines 225-226 and 239).

**Comment 17:** Eq. (9), what is the # of windows for different grids 0.5°x0.5°, 1°x1° and 2°x2°?

**Response 17:** We apologize for not understanding your meaning. May we ask what # represents?

**Comment 18:** L241, "Finally, a global real-time and high-precision $Tm$ lapse rate model was developed and" How can you get the gamma $\gamma$ (the lapse rate of $Tm$) in real-time? How can you get the $TmG$ value at the height of the gridded points from the reanalysis data in real-time? In what applications, the real-time Tm is really needed?

**Response 18:** Thanks for the question you raised. Eq. (9) can be used to calculate the gamma $\gamma$ (the lapse rate of Tm) in real-time. The use of Eq. (9) only requires the input of the day of the year (DOY), so it can achieve real-time calculation for $\gamma$. For example, entering today's DOY can calculate today's $\gamma$.

In addition, obtaining real-time Tm at the height of gridded point requires Eq. (11) and (12). The use of these two equations only requires input of the hour of the day (HOD) and the day of the year (DOY), so it can achieve real-time calculation for Tm. The integration of reanalysis data to obtain the Tm at the height of gridded point cannot be achieved in real-time because the release of reanalysis data has a time delay.

Finally, real-time Tm is required in the application of extreme weather forecast such as rainstorm.

**Comment 19:** L267, "…daily cycle amplitude, and semidaily cycle amplitude at all grid points using the least-squares adjustment using surface-level gridded $Tm$ data calculated from all the ERA5 reanalysis data" In L210, you wrote "Since the daily variation in the lapse rate of $Tm$ can be overshadowed by the annual and semiannual variations." Thus you didn't model the daily cycle amplitude, or semidaily cycle amplitude at all grid points in the Section 3.2. I can't understand why you bring up the daily cycle amplitude, and semidaily cycle amplitude in this Section 4.1.

**Response 19:** Thanks for the question you raised. The NGGTm-H model was developed in Section 3, which can calculate the lapse rate of Tm ($\gamma$). The NGGTm model was developed in Section 4, which can directly calculate Tm. The research objects in Section 3 and 4 are different. The research objects in Section 3 and 4 are the $\gamma$ and Tm, respectively. The statement that "daily variation may be overshadowed by annual and semiannual variations" in Section 3.1 refers to the $\gamma$ rather than Tm. According to Fig. (5) and reference (Sun et al., 2019), it is necessary to consider the daily variation of Tm. The relationship between Sections 3 and 4 is as follows: the NGGTm-H model in Section 3 (composed of Eq. (9) and (10)) is part of the NGGTm model in Section 4 (composed of Eq. (9), (10), (11) and (12)). The NGGTm model is the final model of this study.

Sun, Z. Y., Zhang, B., and Yao, Y. B.: An ERA5-based model for estimating tropospheric delay and weighted mean temperature over China with improved spatiotemporal resolutions, Earth Space Sci., 6, 1926–1941, https://doi.org/10.1029/2019EA000701, 2019.

**Comment 20:** In Eq. (9) of Section 3.2, you estimated the annual cycle amplitude and semiannual cycle amplitude of the lapse rate gamma $\gamma$ of $Tm$. However in the Figure 4 of Section 4.1, you showed the annual cycle amplitude and semiannual cycle amplitude of Tm. I know the lapse rate gamma $\gamma$ of $Tm$ is closely related to Tm. But they are not the same thing. You need to state clearly in the submission what you want to study: lapse rate or Tm.

**Response 20:** Thank you for pointing this out. We agree with this comment. Indeed, our description makes it difficult for readers to understand the relationship between $\gamma$ and Tm. We have added some statements at the beginning of Section 4.1 (see lines 271-273). In addition, the steps for calculating the Tm at user's location using the NGGTm model have been explained in detail (see lines 302-308). Thank you again for your suggestion. Your suggestion made us think deeply and realize that we should explain clearly the writing ideas of the article from the perspective of the readers.

**Comment 21:** L276, "In summary, $Tm$ not only undergoes significant annual and semiannual variations but also experiences significant daily and semidiurnal variation." Again, it is quite perplexing to me that you stated that significant daily and semidiurnal variations here but you didn't study it in Section 3.2.

**Response 21:** Thanks for the question you raised. As mentioned in Response 19, the research subjects in Sections 3 and 4 are different. The research subjects in Section 3 and 4 are $\gamma$ and Tm, respectively. In Section 3, we only consider the annual and semiannual variations of $\gamma$, because its daily variation may be overshadowed by annual and semiannual variations . In Section

4, we consider the daily variation of Tm because it cannot be ignored according to Fig. 6 and reference (Sun et al., 2019).

Sun, Z. Y., Zhang, B., and Yao, Y. B.: An ERA5-based model for estimating tropospheric delay and weighted mean temperature over China with improved spatiotemporal resolutions, Earth Space Sci., 6, 1926–1941, https://doi.org/10.1029/2019EA000701, 2019.

**Comment 22:** L282, "Since the significant variations in the horizontal direction of $Tm$ compared to lapse rate, the estimation of $Tm$ at the gridded points did not use the sliding window algorithm." It is hard to understand. Rephrase it.

**Response 22:** Thanks for your valuable suggestion. We have rephrased this sentence. In the revised manuscript, the relevant content has been modified as follows: "Since the significant variations in the horizontal direction of Tm compared to lapse rate according to Fig. 5 (a) and Fig. 6 (a), it is necessary to develop surface Tm models at each gridded point instead of using sliding windows" (see lines 291-293). As shown in Fig. 5 (a), the annual average value of $\gamma$ is approximately -6 K/km in Qinghai Tibet Plateau with the high-altitude and -5 K/km in eastern China with the low altitude. The difference between them is approximately 1 K/km, which means that altitude variation of 1 km leads to a difference in Tm variation of 1 K. As shown in Fig. 6 (a), the annual average value of Tm is approximately 260 K in Qinghai Tibet Plateau with the high-altitude and 280 K in eastern China with the low altitude. The difference between them is approximately 20 K. This indicates that the significant variations in the horizontal direction of Tm compared to lapse rate. Therefore, it is necessary to develop surface Tm models at each gridded point instead of using sliding windows.

**Comment 23:** This submission has an unusually high frequency of self-citation (e.g. Huang, L. K., Yao, Y. B.).

**Response 23:** We apologize for this question you raised. Due to the significant reference value of the Tm research created by scholars Huang, L. K. and Yao, Y. B., we frequently cited their articles. To avoid any doubts from readers, we have reduced the frequency of citations in their articles.

---

## Referee Report (RR1)

The authors attempted to create a model for temperature lapse rates by establishing a linear relationship between temperature and elevation in near-Earth space. Through this model, users can calculate the corrected temperature value for a specific position and day of the year by providing coordinate information. The NGGTm model was developed by considering the detailed temporal characteristics of temperature. Global radiosonde data and ERA5 reanalysis data verified its accuracy and applicability. An improvement was found in the NGGTm model compared to the Bevis and GPT3 models. Precision and stability improvements have also improved when bias and RMSE ranges are considered.

To increase confidence in the results, this study requires further extension involving the application of the model to GNSS PWV retrieval tasks. However, as a first attempt, I recommend the article for publication.

**Major comment:**

The comment aligns similarly to the Anonymous Referee #2, which remains unanswered.

In line 125, the author mentions: "The ERA5 gridded data from 2012 to 2017 and the radiosonde data in 2017 on the global scale were used to analyze and develop the model in this study."
Furthermore, in line 446, the author mentions: "In general, the NGGTm model can provide real-time and high-precision Tm information without requiring the input of measured meteorological parameters at the global scale."

What does the author state by "real-time" here? Should the model be corrected by a dataset (preferably observational data or ERA5) before a real-time application of the model?

**Minor Comments**

Line 21
Put a comma: "0.50 K, 0.18 K, and 0.06 K"

Line 33
Please change "uneven" to "unevenly".

Line 47
Add the to "the zenith wet delay (ZWD)"; also add hyphen to high-precision.

Line 48
The words "the basic" is inappropriate here. Maybe change to "a primary"?

Line 52
Change "obtain an accuracy" to "obtain accuracy".

Line 54
Remove "by"

Line 62
Change "To improve.." to "In order to improve.."

Line 68
Remove "that is".

Line 76
It should be "a neural network.."

Line 79

Please modify: "The above models have achieved good results when providing the required measured meteorological parameters. However, most of the GNSS stations in the world do not have supporting meteorological sensors installed leading to difficulty in measuring meteorological parameters in real-time."

Line 81
Modify the sentence. Maybe "Therefore, these models are challenging to apply in real-time or near-real-time GNSS PWV detection tasks."?

Line 88
The sentence is not clear.
"Taking the lapse rate into account can not only improve the $Tm$ model accuracy, but also showed significant performances in regions with undulating terrain (Huang et al., 2023b; Sun et al., 2021; Yao et al., 2018)."
Showed→show; What performance? Is it an improvement? Please clarify.

Line 94
Greatly→ significantly

Line 95
Certain → Maybe "specific"?

Line 98
Change "have been applied to" to "has been applied in"

Line 101
Change "developed jointing" to "developed by jointing"

Line 103
Modify the sentence to: "Although the GPT3 model is currently the most representative empirical model with a high precision on the global scale, it does not take into account elevation correction or detailed $Tm$ lapse rate."

Line 110
Add "then"? →".. $Tm$ and then developed a new global grid lapse rate model.."

Section 2.2: Please cite the literature to the mathematical equations for those used in previous research.

Line 214
Change "presents" to "present".

Line 424
Please change the sentence to: "The NGGTm-H model achieved excellent results in the precision verification performed by combining ERA5 reanalysis data and radiosonde data that were not involved in the modeling process."

---

## Author Response (AR2)

**Response to Editor**

Dear editor,

Thank you for offering us an opportunity to improve the quality of our submitted manuscript titled "**A hybrid-grid global model for the estimation of atmospheric weighted mean temperature considering time-varying lapse rate in GNSS precipitable water vapor retrieval**" (gmd-2024-21). We appreciated very much your constructive and insightful comments. In the following, we include a point-by-point response to the comments. We have checked our manuscript carefully for typos, missing co-authors and their affiliations, terminology, updates of data in tables, or updates of variables in equations. In the revised manuscript, all the changes have been highlighted in red. We hope the revised manuscript has now met the publication standard of your journal.

**Comment 1:** Please clarify how the NGGTm model can be applied at "real-time" since the ERA5 data will not be available.

**Response 1:** Thanks for your comment. ERA5 data is really not available in "real-time". But the purpose of this study is to develop an empirical model, which uses historical data to model. When applying the model, it only needs to determine the user's spatial position and input time to estimate $T_m$ without inputting any parameters of ERA5. The workflow of developing the "real-time" $T_m$ model is shown in Figure 1. As mentioned in Section 1, $T_m$ models are divided into meteorological and nonmeteorological parameter models. The meteorological parameter model requires input of "real-time" meteorological parameters, such as Bevis model ($T_m$=70.2+0.72$T_s$). Therefore, such models cannot achieve "real-time" performance. However, nonmeteorological parameter model do not require input of "real-time" meteorological parameters. The $T_m$ model proposed in this study belongs to this type of model and can achieve "real-time" calculation and prediction of the future. The proposed $T_m$ model consists of the following four Equations, which do not require the input of "real-time" meteorological parameters but only time.

$$\gamma^i = A_0{}^i + A_1{}^i \cos\left(2\pi \frac{DOY}{365.25}\right) + A_2{}^i \sin\left(2\pi \frac{DOY}{365.25}\right) + A_3{}^i \cos\left(4\pi \frac{DOY}{365.25}\right) + A_4{}^i \sin\left(4\pi \frac{DOY}{365.25}\right) \quad (1)$$

where $i$ is the number of windows; $\gamma^i$ is the lapse rate of $T_m$ in the ith window; $A_0{}^i$ is the annual mean value of the lapse rate of $T_m$ in the ith window; $(A_1{}^i, A_2{}^i)$ is the annual cycle coefficient of the lapse rate of $T_m$ in the ith window; $(A_3{}^i, A_4{}^i)$ is the semiannual cycle coefficient of the lapse rate of $T_m$ in the ith window; and $DOY$ is the day of the year.

$$T_m^U = T_m^G - \gamma^i(H^U - H^G) \quad (2)$$

where $T_m^U$ is the $T_m$ value at the user height; $T_m^G$ is the $T_m$ value at the height of the gridded points from the reanalysis data; $\gamma^i$ is the lapse rate of $T_m$ at the window where the user is located; $H^U$ is the elevation of the user; and $H^G$ is the elevation of the gridded point from the reanalysis data.

$$T_m^G = B_0 + B_1 \cos\left(2\pi \frac{HOD}{24}\right) + B_2 \sin\left(2\pi \frac{HOD}{24}\right) + B_3 \cos\left(4\pi \frac{HOD}{24}\right) + B_4 \sin\left(4\pi \frac{HOD}{24}\right) \quad (3)$$

$$B_i = b_{i0} + b_{i1}\cos\left(2\pi\frac{DOY}{365.25}\right) + b_{i2}\sin\left(2\pi\frac{DOY}{365.25}\right) + b_{i3}\cos\left(4\pi\frac{DOY}{365.25}\right) + b_{i4}\sin\left(4\pi\frac{DOY}{365.25}\right) \quad (4)$$

where $T_m^G$ is the $T_m$ value at the gridded points; $B_i$ is the daily variation coefficient of $T_m$; and $HOD$ is the UTC time. After Eq. (4) was used to expand Eq. (3), $b_{ij}$ (i=0,1,2,3,4 and j=0,1,2,3,4), which represents the 25 coefficient terms of the model, was calculated. $DOY$ is the day of the year.

[Figure]

**Figure 1 The workflow of developing the proposed $T_m$ model.**

**Comment 2:** What kind of data will be used to represent the temporal characteristics of "real-time" temperature, an important advantage of applying the NGGTm model to GNSS PWV retrieval tasks?

**Response 2:** Thanks for the question you raised. Although ERA5 data cannot be available in "real-time", we use historical ERA5 data to analyze the temporal variation characteristics of $T_m$ and achieve "real-time" modeling through periodic functions. The proposed $T_m$ model only requires inputs of day of year (DOY) and hour of day (HOD).

---

## Author Response (AR3)

**Response to Editor and Reviewers**

Dear editor and reviewers,

Thank you for offering us an opportunity to improve the quality of our submitted manuscript titled "**A hybrid-grid global model for the estimation of atmospheric weighted mean temperature considering time-varying lapse rate in GNSS precipitable water vapor retrieval**" (gmd-2024-21). We appreciated very much your constructive and insightful comments. In the following, we include a point-by-point response to the comments. We have had a native English speaker proofread this manuscript and carefully correct the grammar errors. Unclear sentences and phrases of this manuscript have been checked and corrected. In the revised manuscript, all the changes have been highlighted in red. We hope the revised manuscript has now met the publication standard of your journal.

**Comment 1:** The phrase of "surface grid Tm" is very confusing. Please reword this phrase.

**Response 1:** Thank you for pointing this out. All the phrase of "surface grid Tm" in the manuscript have been changed to "surface gridded Tm data" (see L19).

**Comment 2:** L21: higher? I think this is a typo. This should be lower?

**Response 2:** Yes, your point is correct. This should be lower (see L21).

**Comment 3:** L51: This sentence is confusing. Please revise it. What did the authors mean by "obtaining the ZTD information by integrating atmospheric reanalysis data"? Why is it related to high-precision observation data provided by the GNSS base station network?

**Response 3:** Thanks for your comment. We have cerrected this sentence to "high-precision ZTD information can be obtained through data processing with high-precision GNSS data processing software". What we want to express is ZTD can be obtained from GNSS rather than atmospheric reanalysis data (see L52).

**Comment 4:** L90: Please change significant performances to significant improvement.

**Response 4:** Thanks for your suggestion. " significant performances" has been cerrected to "significant improvement" (see L90).

**Comment 5:** L102: Please define the GPT-series models first.

**Response 5:** Thank you for pointing this out. We have defined the GPT-series models on L101.

**Comment 6:** Please revise the sentences from L102 to L106. Why did the authors mention the GPT2w model? What are the differences between GPT2w and GPT3 models?

**Response 6:** Thanks for the question you raised. To avoid misleading readers, we have revised the sentences by removing the introduction of GPT2w because what we want to emphasize is the GPT3 model. Global pressure and temperature (GPT) series models which include GPT, GPT2, GPT2w and GPT3 model. GPT3 is the latest generation model (see L101 to L104).

**Comment 7:** L105: elevation correction for what?

**Response 7:** Thanks for the question you raised. The purpose of elevation correction is to reduce interpolation errors. The detailed explanation can be refered from L162 to L165. In order to make it easier for readers to understand, We have revised "elevation correction" to "vertical correction" (see L104).

**Comment 8:** L115: Please revise this sentence. "Tm data from radiosonde stations with ERA5 reanalysis data"?

**Response 8:** Thank you for pointing this out. Whave revised this sentence to "the NGGTm model was compared with the Bevis and GPT3 models using $T_m$ data from radiosonde stations and ERA5 reanalysis data" (see L113).

**Comment 9:** L134: H in Eq. (1) is the integral range? It should be an integration variable. I would rewrite Eq. (1) in a clearer way.

**Response 9:** Thanks for your suggestion. We have rewrited Eq. (1) and explained the meaning of each variable in a clearer way (see L132 to L134).

**Comment 10:** Please revise L161. I believe the authors meant the disparities between the elevation of the analysis at the GNSS station and the actual elevation of the GNSS station.

**Response 10:** Thanks for your comment. We have revised "user point" to "target point" (see L161). The task of the Tm model is to calculate the Tm value at any spatial position. It should be noted that radiosonde stations can provide Tm instead of GNSS stations. The Tm derived from radiosonde stations is used as a reference value to validate the accuracy of the model. In practical applications, the Tm model can calculate Tm values including but not limited to radiosonde station locations.

**Comment 11:** Tm is a weighted mean temperature. Any phrases like "$T$m elevation" or "vertical $T\ m$ information" are very confusing.

**Response 11:** Thank you for pointing this out. We have revised them to "Tm vertical cerrection" uniformly (see L89 and L104).

**Comment 12:** L164: elevation of what?

**Response 12:** Thanks for the question you raised. In this sentence, the object of variation is Tm rather than elevation, so we revised this sentence to "the vertical $T_m$ variation is much larger than the variation in the horizontal direction" (see L161).

**Comment 13:** L165: what did the authors mean by "with elevation in depth"? with different elevations?

**Response 13:** Thanks for the question you raised. The meaning what we want to express is "to further analyze the variation in $T_m$ with elevation". The revised sentence can be seen L164.

**Comment 14:** L262: Did the authors mean "Tm from radio stations as the reference" and "the Tm calculated by the ERA5 surface-level data". Please revise this sentence.

**Response 14:** Thank you for pointing this out. We have revised this sentence to "the precision statistics obtained for the three resolutions of the NGGTM-H model tested using $T_m$ data from global radiosonde stations in 2017" (see L256). We validate the accuracy of the NGGTM-H model using radiosonde station data instead of ERA5 data. ERA5 data is only the starting value for the NGGTM-H model.

**Comment 15:** L263-264: This sentence is very confusing. " what is comparing to the reference data"? Compared to the Tm from radiosonde stations, the correction made by the NGGTm-H model is too large in the land areas but too small in marine areas?

**Response 15:** Thanks for the question you raised. Reference data is the Tm from radiosonde stations. The Tm calculated using NGGTm-H model is compared to the reference data. ERA5 data is only the starting value for the NGGTM-H model. We have revised this sentence to "positive mean biases with relatively small absolute values were obtained for the NGGTm-H model at the three resolutions taking $T_m$ data from radiosonde stations as reference values" (see L260 to L262).

In addition, we have also checked and revised other errors. Please refer to the manuscript for details.Thanks again for your constructive and insightful comments.

---

## Author Response (AR4)

**Response to Editor and Reviewers**

Dear editor and reviewers,

Thank you for offering us an opportunity to improve the quality of our submitted manuscript titled "**A hybrid-grid global model for the estimation of atmospheric weighted mean temperature considering time-varying lapse rate in GNSS precipitable water vapor retrieval**" (gmd-2024-21). We appreciated very much your constructive and insightful comments. In the following, we include a point-by-point response to the comments. We have improved the English writing of this manuscript. Unclear phrases and related sentences have been revised, particularly the sections for analyzing and constructing a model for the Tm lapse rate. In the revised manuscript, all the changes have been highlighted in red. We hope the revised manuscript has now met the publication standard of your journal.

**Comment 1:** A major issue is the confusing description of Tm, which is defined as a vertically weighted mean temperature, i.e., no vertical dependence. The vertical correction is included to consider the varying topography height to better reflect hbot in Eq. (1), isn't it? However, many sentences refer to Tm as a "vertical dependent" variable. These sentences must be clarified to deliver the concept of "adjusting the given Tm at starting height to the target height" (line 278). I only listed the sentences I found, but please carefully revise all the sentences with such an issue. Can the authors avoid using "elevation of Tm"? Also, "user height", "target point" and "gridded point" need clear definitions and are used consistently. Is the elevation of a target point the same as the altitude of the GNSS ZTD receiver? Since this study inverts PWV with GNSS ZWD information, there should be a clear sentence that defines how to consider the altitude of the ground-based GNSS receiver.

- Line 166: vertical Tm variation? Since Tm is a vertically weighted mean temperature, there is no vertical dependence. Please revise this sentence.

- Line 167: vertical Tm information.

- Line 168: layered Tm data and elevation of Tm. Is the "elevation" here the same as the hbot in Eq. (1)?

- Line 176: The term "grid-level" is constantly used, but it is not well defined. Is this referred to the surface grid?

- Same for the caption of Fig. 1.

- Line 243: Is the height in this sentence referred to as the height of the surface grid? Same question for the elevation.

- Line 278: surface Tm model, whose hbot is at the surface?

**Response 1:** Thanks for your suggestion. We agree with you. We have revised the sentences you mentioned and other unclear sentences. The phrase "elevation of Tm" have been changed to "Tm height". The words "user" and "target" have been unified as "target". In this study, the height of a target point is the same as the altitude of the GNSS ZTD receiver. We have explained the "target height" (see line 166).

- Line 166: The phrase "vertical Tm variation" has been changed to "Tm variation in the vertical direction" (see line 165).

- Line 167: We have cerrected this sentence to "it is necessary to correct Tm vertically" (see lines 166).

- Line 168: The "elevation" here is the same as the $h_{bot}$ in Eq. (1). The phrase "between the layered Tm data and elevation of Tm" has been changed to "between the layered Tm data and corresponding height" (see line 167).

- Line 176: The term "grid-level" is not the "surface grid". "grid-level" has been changed to "layered" (see line 169, 174 and 175).

- Line 243: The word "elevation" has been changed to "height". $H^T$ is the height of the target point; and $H^G$ is the height of the gridded point (see line 241).

- Line 278: Yes, the height of surface Tm model is at the surface. This sentence have been cerrected to "it is necessary to develop a Tm model whose height is at the surface (named as surface Tm model)" (see line 274).

**Comment 2:** Line 48-58: The relationship between PWV and ZWD should be better structured. Maybe starting with a sentence like "The inversion of PWV uses the wet component of the zenith total delay", followed by more explanations about ZTD.

**Response 2:** Thank you for pointing this out. The relationship between PWV and ZWD has been structured to "PWV can be inverted by multiplying the wet component of the zenith total delay (ZTD) with the water vapor conversion factor. The ZTD consists of two parts: the zenith hydrostatic delay (ZHD) and the zenith wet delay (ZWD)" (see lines 47-49).

**Comment 3:** A brief paragraph should be included at the end of the introduction to explain the structure of this paper.

**Response 3:** Thanks for your suggestion. We have explained the structure of this paper at the end of the introduction (see lines 114-117).

**Comment 4:** Line 93: Since Tm is a weighted mean temperature, how can it be corrected to a certain height? Please revise this sentence. I think you want to say "correcting Tm to consider the varying topography height".

**Response 4:** Thank you for pointing this out. We have revised this sentence to "the Tm lapse rate is an effective means of correcting Tm to consider the varying topography height" (see line 91).

**Comment 5:** Lines 218-219: What did the authors mean by "poor performance in spatial difference" based on Fig.3?

**Response 5:** Thanks for the question you raised. What we want to emphasize is "the variation law of the lapse rate of Tm differs spatially" rather than "poor performance in spatial difference". We have revised this sentence to "the above analysis demonstrated that the variation law of the lapse rate of Tm differs spatially. This makes it difficult to accurately grasp the variation law of the lapse rate of Tm in developing a global uniform model for the lapse rate of Tm" (see lines 216-217).

**Comment 6:** First paragraph on Page 11: I assume the ERA5 analysis is used to construct the coefficients in Eq. 9 with least square fitting. If that's correct, please clearly specify the statement. Once the coefficients are determined, the model can be executed in real-time. The ability to execute the model at real time should be emphasized in the summary.

**Response 6:** Thanks for your suggestion. We have clearly specified the data used to calculate the five coefficients (see line 246). In addition, the ability to execute the model at real time has be emphasized in the summary (see line 437).

**Comment 7:** Line 400: It is difficult to tell that Fig. 9d has smaller RMSEs than Figs. 9a-9c! Please double-check the arrangement of the subplots.

**Response 7:** Thank you for pointing this out. The inconsistent color bar range makes it difficult to compare for model performance. The color bar range has been adjusted to be consistent (see line 387).

In addition, we have also checked and revised other phrases and related sentences. Please refer to the revised manuscript for details.Thanks again for your constructive and insightful comments.

---

## Author Response (AR5)

**Response to Editor and Reviewers**

Dear editor and reviewers,

Thank you for offering us an opportunity to improve the quality of our submitted manuscript titled "**A hybrid-grid global model for the estimation of atmospheric weighted mean temperature considering time-varying vertical adjustment rate in GNSS precipitable water vapor retrieval**" (gmd-2024-21). We appreciated very much your constructive and insightful comments. In the following, we include a point-by-point response to the comments. We hope the revised manuscript has now met the publication standard of your journal. Please note that the phrase "lapse rate" has been changed to "vertical adjustment rate" in the title.

**Comment 1:** I strongly suggest that the authors should not use the phrase "Tm lapse rate". If you are referring to time-varying or horizontally-varying changes in Tm, that is ok. But you apply the vertical adjustment to consider the altitude difference. I would suggest naming gamma as the vertical adjustment rate.

- Line 166: Again, Tm is a vertically integrated value. There is no Tm variation in "vertical direction".

**Response 1:** Thanks for your suggestion. We have named gamma as the vertical adjustment rate.

- Line 166: To avoid confusing readers, we have deleted this sentence "the Tm variation in the vertical direction is much larger than that in the horizontal direction". Deleting this sentence will not change the meaning of the paragraph.

**Comment 2:** The authors emphasize the feasibility of the NGGTm model in calculating Tm in real-time, but there are still sentences that are confusing to the readers.

- Line 244: Tm_G uses reanalysis data. I assume that this is for subsection 3.3(?). As the authors introduce section 4, Tm_G should be calculated by Eqs. (11) and (12). Please provide clarification about the use of reanalysis data for Tm_G.

**Response 2:** Thank you for pointing this out. We have revised the sentences you mentioned.

- Line 244: Yes, this is for section 3.3. We have clarified the use of reanalysis data for Tm_S, note that Tm_G has been changed to Tm_S.

**Comment 3:** Confusing sentences about the definition of "height". As the authors have defined Tm, please refer to h_bot when talking about "height" or surface. If the authors want to use "layered" Tm data, please provide a clear definition, such as the h_bot starting at different heights.

- Line 278: "It is necessary to develop a surface Tm model whose height is at the surface". Did the authors mean h_bot is now surface (e.q, zero height)?

- Line 296: "Therefore, a new hybrid-grid global Tm model considering time-varying lapse rate was developed on the basis of the NGGTm-H1 model, which used surface data of ERA5 reanalysis recorded from 2012 to 2016". Did the authors mean the "NGGTm-H model" used the surface data of ERA5 reanalysis or the new hybrid-grid Tm model? In the authors' response to my previous comments, the authors specifically clarified that the "grid-level" is "NOT" the surface. Please rewrite this sentence.

- Line 312: "Eq. (11) and (12) to calculate the Tm at the height of the grid points". I assume that Eqs. (11) and (12) are for the Tm whose h_bot are at the surface. If yes, please revise this sentence (height of the grid points -> surface).

    **Response 3:** Thank you for pointing this out. We have referred to h_bot when talking about "height" or surface and provided a clear definition for "layered" Tm data.

- Line 278: Yes, h_bot is now surface. We have revised the word "height" to "h_bot" (see line 295).

- Line 296: The new hybrid-grid Tm model used the surface data of ERA5 reanalysis. We have revised this sentence to "therefore, a new global Tm model considering time-varying vertical adjustment rate was developed which used the integrated surface Tm of ERA5 reanalysis recorded from 2012 to 2016 on the basis of NGGTm-H1 model".

- Line 312: Yes, Eqs. (11) and (12) are for the Tm whose h_bot are at the surface. We have revised the phrase "height of the grid points" to the word "surface".

    **Comment 4:** In conclusion, please add one sentence to emphasize the concept of "hybrid grid". The term "hybrid grid" is only mentioned in the title but not adequately explained in the text (e.g., line 295).

    **Response 4:** Thanks for your suggestion. We have added one sentence to emphasize the concept of "hybrid grid".

**Minor:**

    **Comment 5:** Line 298: I don't see significant horizontal variations in Figs 5a and 6a. Please make sure this is not a typo.

**Response 5:** Thank you for pointing this out. This is indeed a typo. We have revised "Fig. 6 (a)" to "Fig. 3 (a)".

**Comment 6:** I don't know why the number model coefficient is 10. Did the authors mean 100 (25 coefficients x 4 surrounding grids)?

**Response 6:** Thank you for pointing this out. The number of model coefficients is 100 (25 coefficients x 4 surrounding grids) instead of 10.

Thanks again for your constructive and insightful comments.